# Inhalation of the prodrug PI3K inhibitor CL27c improves lung function in asthma and fibrosis

Carlo C. Campa[1], Rangel L. Silva[1,2], Jean P. Margaria[1], Tracey Pirali [3], Matheus S. Mattos[4], Lucas R. Kraemer[4], Diego C. Reis[4,5], Giorgio Grosa[3], Francesca Copperi[1], Eduardo M. Dalmarco[6], Roberto C.P. Lima-Júnior[1,7], Silvio Aprile[3], Valentina Sala [1], Federica Dal Bello [8], Douglas Silva Prado[2], Jose Carlos Alves-Filho [2], Claudio Medana[8], Geovanni D. Cassali[5], Gian Cesare Tron[9,10], Mauro M. Teixeira[11], Elisa Ciraolo [1,12], Remo C. Russo[4,11] & Emilio Hirsch [1,10]

PI3K activation plays a central role in the development of pulmonary inflammation and tissue remodeling. PI3K inhibitors may thus offer an improved therapeutic opportunity to treat non-resolving lung inflammation but their action is limited by unwanted on-target systemic toxicity. Here we present CL27c, a prodrug pan-PI3K inhibitor designed for local therapy, and investigate whether inhaled CL27c is effective in asthma and pulmonary fibrosis. Mice inhaling CL27c show reduced insulin-evoked Akt phosphorylation in lungs, but no change in other tissues and no increase in blood glycaemia, in line with a local action. In murine models of acute or glucocorticoid-resistant neutrophilic asthma, inhaled CL27c reduces inflammation and improves lung function. Finally, inhaled CL27c administered in a therapeutic setting protects from bleomycin-induced lung fibrosis, ultimately leading to significantly improved survival. Therefore, local delivery of a pan-PI3K inhibitor prodrug reduces systemic on-target side effects but effectively treats asthma and irreversible pulmonary fibrosis.

[1] Department of Molecular Biotechnology and Health Sciences, Molecular Biotechnology Center, University of Torino, Via Nizza 52, 10126 Torino, Italy. [2] Department of Pharmacology, Ribeirão Preto Medical School, University of São Paulo, Avenida Bandeirantes 3900, 14049-900 Ribeirao Preto, Brazil. [3] Dipartimento di Scienze del Farmaco, Università degli Studi del Piemonte Orientale "A. Avogadro", Largo Donegani 2, 28100 Novara, Italy. [4] Laboratory of Pulmonary Immunology and Mechanics, Department of Physiology and Biophysics, Institute of Biological Sciences, Universidade Federal de Minas Gerais/UFMG, Avenida Antonio Carlos 6627, Belo Horizonte 31270-901, Brazil. [5] Laboratory of Comparative Pathology, Department of General Pathology Institute of Biological Sciences, Universidade Federal de Minas Gerais/UFMG, Avenida Antonio Carlos 6627, Belo Horizonte 31270-901, Brazil. [6] Department of Clinical Analysis, Universidade Federal de Santa Catarina/UFSC, Rua Delfino Conti, S/N, Florianopolis 88040-370, Brazil. [7] Laboratory of Pharmacology of Inflammation and Cancer, Department of Physiology and Pharmacology, Universidade Federal do Ceará/UFC, Rua Cel Nunes de Melo 1127, Fortaleza 60430-270, Brazil. [8] Department of Molecular Biotechnology and Health Sciences, Mass Spectrometry Unit, University of Torino, Via Giuria 5, 10125 Torino, Italy. [9] Dipartimento di Scienze del Farmaco, Università degli Studi del Piemonte Orientale "A. Avogadro", largo Donegani 2, 28100 Novara, Italy. [10] Kither Biotech S.r.l., Via Nizza 52, 10126 Torino, Italy. [11] Laboratory of Immunopharmacology, Department of Biochemistry and Immunology, Institute of Biological Sciences, Universidade Federal de Minas Gerais/UFMG, Avenida Antonio Carlos 6627, Belo Horizonte 31270-901, Brazil. [12] Max Delbrück Center for Molecular Medicine, 13125 Berlin, Germany. These authors contributed equally: Carlo C. Campa, Rangel L. Silva, Elisa Ciraolo, Remo C. Russo, Emilio Hirsch. Correspondence and requests for materials should be addressed to E.C. (email: Elisa.Ciraolo@mdc-berlin.de)

Asthma, a persisting and recurring inflammation of the respiratory tract, represents a common chronic disease and a growing public health problem worldwide[1]. Corticosteroids are the mainstay treatment but as high as 10% of "severe asthma" patients show resistance to therapy[2]. While chronic inflammation in asthma can represent a potential pathogenic mechanism of airway fibrosis, etiology of other pulmonary fibrotic conditions, like Idiopathic Pulmonary Fibrosis (IPF), is still unknown[3]. Nonetheless, the involvement of the phosphoinositide 3-kinase (PI3K) pathway in the aberrant activation of immune cells, as well as pulmonary fibroblasts, has suggested PI3K enzymes as potential therapeutic targets in both severe asthma and IPF[4–6].

Mammalian class I PI3Ks constitute a family of homologous kinases, comprising the four enzymes PI3Kα, PI3Kβ, PI3Kγ, and PI3Kδ. These proteins are lipid kinases that synthetize the membrane anchored second messenger phosphatidylinositol 3,4,5-trisphosphate $[\text{PtdIns}(3,4,5)P3]$[7], which in turn triggers downstream effectors, like the serine/threonine kinase Akt/PKB, eventually evoking a complex signaling network[7]. Functional studies show that the four class I PI3Ks play a key role in the regulation of inflammation and fibrosis[8]. For example, PI3Kγ and δ control various leukocyte functions, including proliferation, differentiation, migration, and survival[8] In addition, PI3Kα and β control proliferation of various lung cell types and at least two isoforms, PI3Kγ and β are involved in pulmonary inflammation and fibrotic remodeling, as their inhibition is protective in models of asthma and lung fibrosis[9–11]. Although each class I PI3K appears prominently associated to a specific set of functions, redundancy within the family is frequent[7,12,13]. Therefore, pan-PI3K inhibitors are expected to maximize therapeutic efficacy but their use is severely restricted by their on-target toxicity[7].

A potential way to overcome side effects in pulmonary diseases, like severe asthma and IPF, is to avoid systemic absorption by the local delivery of inhaled active compounds. A prodrug pan-PI3K inhibitor like CL27c[14] could present an even safer profile but its in vivo efficacy has yet remained unaddressed. To explore the safety and efficacy of this strategy, CL27c was administered through the inhaled route in murine models of asthma, severe asthma and pulmonary fibrosis and found to be able to significantly reduce inflammation, improve lung function, and reduce bleomycin-induced lethality with negligible systemic toxicity.

## Results

**Effects of CL27c on PI3K-dependent cell functions.** Pan-PI3K inhibitors are potentially effective in treating airway inflammation and fibrosis but their systemic side effects limit their use. To circumvent this drawback, CL27c (Supplementary Fig. 1a) (Patent: WO/2012/073184) was designed as a cell-permeable lipophilic ester prodrug[14]. This compound is inactive in PI3K enzymatic assay with purified proteins but, once inside the cytoplasm, is metabolized by unspecific esterases into CL27e (Supplementary Fig. 1a–c), a potent pan-class I PI3K inhibitor, acting at nanomolar concentration[14]. To confirm PI3K-selective inhibition, CL27e was tested against 250 kinases. Notably, CL27e showed a high selectivity for PI3Kα (23 and 81% inhibition at low and high concentrations, respectively), (Supplementary Fig. 2).

In cell-based assays, CL27c readily reduced Akt/PKB phosphorylation in chronic lymphocytic leukemia (CLL) cells[15] stimulated with anti-IgM antibodies (Fig. 1a and Supplementary Fig. 3a), as well as in murine neutrophils and bone marrow-derived macrophages (BMDM) stimulated with C5a (Fig. 1a and Supplementary Fig. 3b, c). Similar results were obtained with human blood mononuclear cells (PBMC) stimulated with CD38/28 (Supplementary Fig. 3d), thus indicating that the prodrug can be effectively converted into the active compound in different cell types.

Chemical features of CL27c indicated that this compound could be suitable for local treatment of inflammation. To test whether CL27c could inhibit typical PI3K-dependent inflammatory responses[16–18], fMLP-triggered respiratory burst of primary human neutrophils was analyzed. As expected, CL27c efficiently and dose-dependently inhibited reactive oxygen species production (Fig. 1b). In a concentration-dependent manner, CL27c was also found to block proliferation of human peripheral blood mononuclear cells (PBMC) stimulated with antibodies against CD28 (Fig. 1c). Remarkably, CL27c dosed at 5 μM displayed an effect comparable to the reference compound dexamethasone. In cell migration assays, CL27c was found to significantly reduce CXCL12- and C5a-induced chemotaxis of murine bone marrow-derived macrophages, thus confirming the inhibition of key PI3K-dependent inflammatory responses (Fig. 1d).

**In vivo and in vitro ADME characterization.** To define whether this prodrug was suitable for inhaled administration, CL27c was studied to evaluate selected ADME (absorption, distribution, metabolism, and excretion) and pharmacokinetic properties. First, stability of CL27c and CL27e was analyzed in mouse plasma (Supplementary Table 1) and in mouse, rat, and human liver microsomes in the presence of nicotinamide adenine dinucleotide phosphate (NADPH; Supplementary Table 2). Next, clearance of CL27c was analyzed in C57BL/6J mice after either intra-venous or oral administration. Peak plasma concentration 5 min after i.v. administration showed that CL27e was twice more abundant than CL27c (Supplementary Table 3), as previously shown[14]. After i.v. administration, a rapid elimination half-life (t½) of 0.65 h, as well as a high clearance (CL) of 191 ml/min/kg swiftly reduced plasma concentration of the two compounds (Supplementary Table 3). Clearance of CL27c and CL27e after oral administration could not be determined due to their low bioavailability (Supplementary Table 3). Next, pharmacokinetics after inhalation was studied in C57BL/6J mice after delivering an aerosolized micro-suspension of CL27c. As shown in Table 1 and Supplementary Fig. 1b, CL27c dose-dependently reached the lungs and its metabolite CL27e was detectable after inhalation of the highest dose. Conversely, in the plasma, CL27c and CL27e were generally undetectable and their traces could only be observed after 1 h of exposure at the highest dose (Table 1, Supplementary Fig. 1c). Altogether, this initial ADME characterization indicates that inhaled CL27c effectively reaches the lungs where it is converted into the active compound CL27e and that both of these chemical entities show negligible systemic absorption. This assumption was hence confirmed in vivo, where treatment with PI3K inhibitors with systemic absorption is known to cause hyperglycemia and insulin insensitivity. Inhaled CL27c was unable to modify glycemia (vehicle 206.0 ± 4.9 mg/dl; CL27c 213.0 ± 5.6 mg/dl; N.S., $n = 4$ per treatment) and did not change the drop in glycemia 10 min after an insulin challenge (vehicle 18.30 ± 2.82% drop; CL27c 17.45 ± 3.00% drop; N.S., $n = 4$ per treatment). In line with the local action of CL27c, lungs of treated mice did not show elevation of protein kinase B (Akt) phosphorylation in Ser473, in response to intraperitoneal administration of insulin (Fig. 1e). Conversely, insulin was able to normally increase Akt phosphorylation in either heart or liver of the same mice (Fig. 1e). Therefore, the inhaled prodrug promotes local effects but prevents systemic absorption of pharmacologically relevant amounts of the active compound.

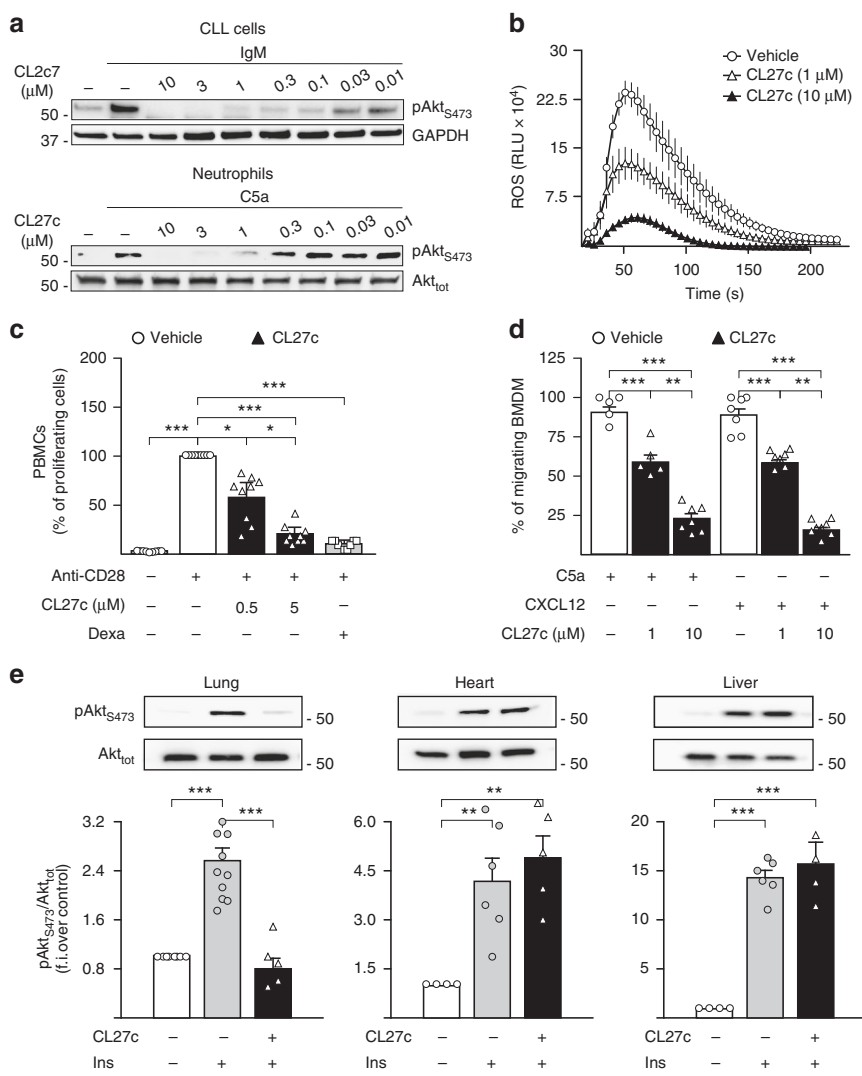

**Fig. 1** CL27c inhibits PI3K class I-mediated effects in vitro and in vivo. **a** Dose response of CL27c (10 nM–10 µM) on Akt phosphorylation in CLL cells (upper panel) and primary murine neutrophils (lower panel), stimulated with IgM (20 µg/ml) and C5a (25 nM), respectively. Western blots are a representative example of at least 5 independent experiments. **b** Inhibitory effect of CL27c (1 and 10 µM) on time-course ROS production by fMLP (2.5 µM)-activated LPS-primed neutrophils (n = 6 independent experiments for each group). Vehicle (0.1% DMSO diluted in PBS solution) was used as a control group. **c** Anti-proliferative effect of CL27c (0.5 and 5 µM) or Dexamethasone (Dexa, 5 mg/kg via i.p. injection) in PBMCs stimulated with anti-CD28 (n = 9 independent experiments for each group). **d** CL27c (1 and 10 µM) inhibits the chemotactic effect of CXCL12 and C5a (100 and 50 nM, respectively) on murine bone marrow-derived monocytes (BMDM) (from left to right, n = 5, 6, 7, 7, 7, 8 independent experiments, respectively). **e** Inhaled CL27c (2 mg/ml suspension/30 min exposure) prevents insulin-dependent (1 UI/kg in PBS i.p.) pAkt activation in the lungs, but neither in the heart nor in liver. Upper panel: representative western blot showing Akt phosphorylation on Ser473 and total Akt. Lower panel: quantification of fold induction relative to the baseline (f.i.) (from left to right, n = 8, 10, 5, 4, 6, 5, 4, 6, 4 independent experiments, respectively). Results represent mean ± s.e.m. *P < 0.05, **P < 0.01, ***P < 0.001 determined using one-way (**c**, **e**) or two-way (**d**) ANOVA followed by Bonferroni post-hoc test

### Table 1 Pharmacokinetics of aerosolized CL27c

| | Hours from exposure | 0.2 mg/ml CL27c | | 2 mg/ml CL27c | |
| --- | --- | --- | --- | --- | --- |
| | | CL27c | CL27e | CL27c | CL27e |
| Lung (ng/g) | 1 | 32.6 ± 5.1 | n.d. | 169.2 ± 17.7 | 4.5 ± 1.0 |
| | 3 | 10.9 ± 0.5 | n.d. | 34.2 ± 5.2 | 0.1 ± 0.1 |
| | 6 | 10.9 ± 1.8 | n.d. | 41.6 ± 6.5 | 0.5 ± 0.2 |
| Plasma (ng/ml) | 1 | n.d. | n.d. | 1.2 ± 0.1 | 1.34 ± 0.62 |
| | 3 | n.d. | n.d. | n.d. | <1 |
| | 6 | n.d. | n.d. | n.d. | <1 |
| Values are the mean of at least three animals. n.d. not determined | | | | | |

**CL27c reduces inflammation in a murine model of asthma**. The finding that CL27c has pharmacokinetic properties suitable for topical administration, prompted to the assessment of CL27c efficacy in murine models of pulmonary inflammation and fibrosis. First, the anti-inflammatory effects of inhaled CL27c were evaluated in an acute model of allergic asthma. CL27c was tested at different doses (0.2, 2, and 20 mg/ml), but the lowest dose had no efficacy on ovalbumin (OVA)-induced acute asthma and the highest dose showed inconsistent outcomes, likely due to solubility problems. Therefore, the intermediate dose (2 mg/ml) was used for all subsequent experiments. OVA-immunized mice were challenged with OVA (Fig. 2a and Supplementary Fig. 4a) and pulmonary inflammation, as well as CL27c target engagement were analyzed on day 29. Treatment with CL27c was clearly accompanied by a decrease in Akt

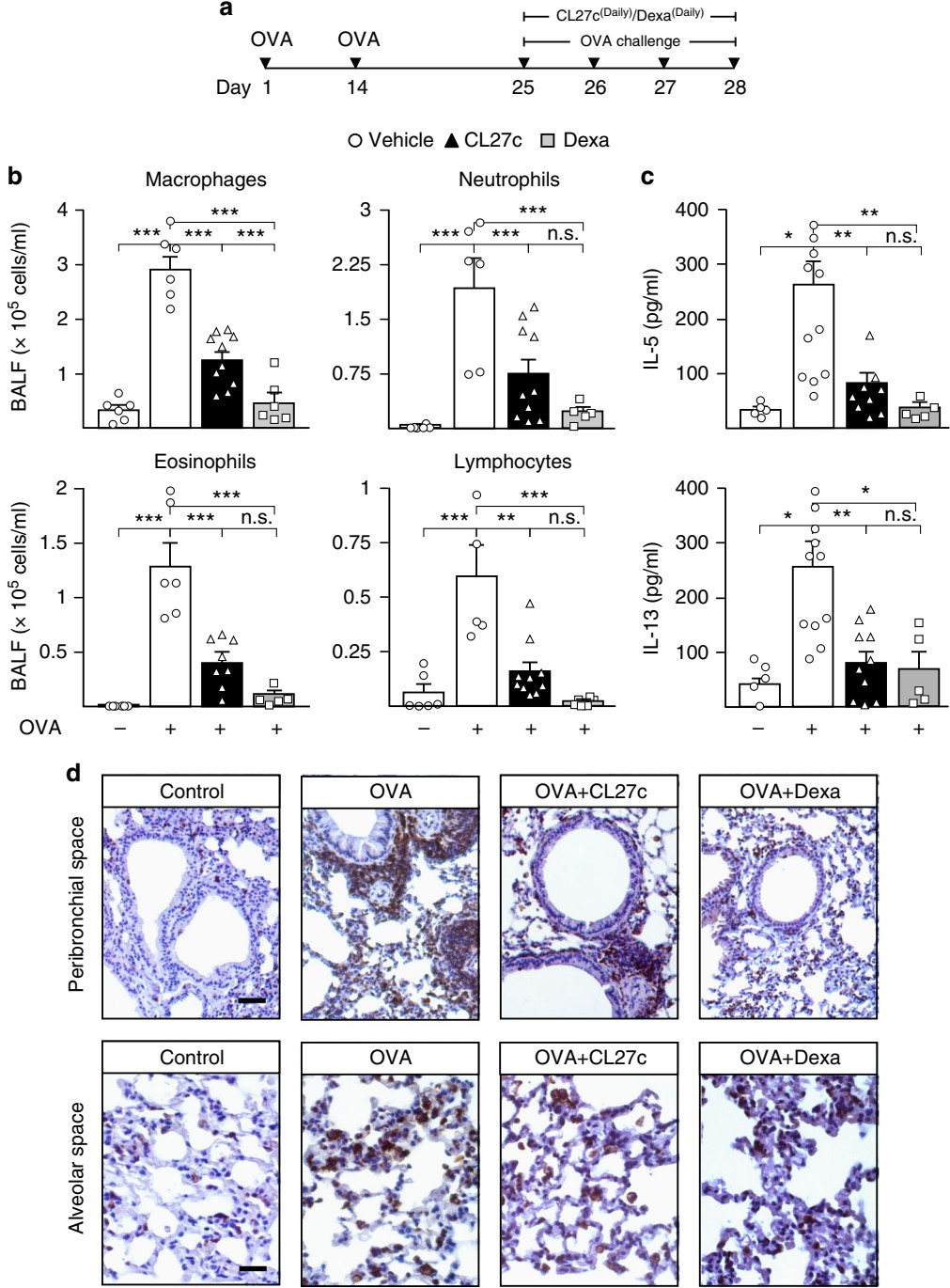

**Fig. 2** CL27c reduces inflammation in a model of OVA-induced acute allergic asthma. **a** Protocol of OVA-induced acute allergic asthma and treatment regimen with inhaled CL27c (2 mg/ml) or Dexamethasone (Dexa, 3 mg/kg, i.p.). **b** Quantification of macrophage, neutrophil, eosinophil, and lymphocyte infiltration in BALF of control and OVA-treated mice in the presence of vehicle (white bars), CL27c (black bars) or Dexa (gray bars) (from left to right, macrofages $n = 6, 6, 10, 6$, neutrophils $n = 6, 6, 10, 5$, eosinophils $n = 6, 6, 8, 5$, and lymphocytes $n = 6, 5, 10, 6$ independent experiments, respectively). **c** Analysis of IL-5 and IL-13 levels in the lung tissue of control and OVA-treated mice in the presence of vehicle, CL27c or Dexa (from left to right IL-5 $n = 5$, 11, 9, 5 and IL-13 $n = 5$, 11, 10, 5 independent experiments, respectively). **d** Histopathological analysis of lung of peribronchial (upper panels) and alveolar (lower panels) spaces showing CD18$^+$ leukocyte infiltration (brown cells) in CL27c or Dexamethasone (Dexa)-treated mice after the OVA challenge. Bar = 50 μm (upper panel) and 10 μM (lower panel). Results represent mean ± s.e.m., *$P < 0.05$, **$P < 0.01$, ***$P < 0.001$ determined using one-way ANOVA followed by Bonferroni post-hoc test. n.s. = non-significant ($P > 0.05$)

phosphorylation in lung lysates and sections (Supplementary Fig. 4b, c). Subsequently, pulmonary inflammation was analyzed and whereas control mice showed a negligible inflammatory infiltrate in the bronchoalveolar lavage fluid (BALF), sensitized animals exposed to vehicle showed a marked BALF accumulation

of inflammatory cells, including macrophages, neutrophils, eosinophils, and lymphocytes (Fig. 2b). Systemic treatment with the reference anti-inflammatory drug dexamethasone significantly reduced this infiltration (Fig. 2b) but inhaled CL27c remarkably reached a similar effect and significantly reduced macrophages

(60%), neutrophils (68%), eosinophils (70%), and lymphocytes (72%) accumulation in BALF (Fig. 2b).

Interleukin 5 (IL-5) and 13 (IL-13) regulate airway obstruction, hyperreactivity, and remodeling[19,20]. As expected, OVA induced a marked 80% increase of either IL-5 or IL-13, respectively (Fig. 2c). Conversely, inhalation of CL27c restored both IL-5 and IL-13 to the levels detected in unsensitized control mice, similarly to the reference compound dexamethasone (Fig. 2c).

In line with these results, analysis by immunohistochemistry of CD18-positive inflammatory cells recruited to the lung showed that OVA triggered the accumulation of leukocytes in the peribronchial spaces (Fig. 2d upper panel), as well as in the alveolar tissue (Fig. 2d lower panel). Conversely, inhaled CL27c reduced leukocyte recruitment (Fig. 2d) thus further supporting the notion that local inhibition of the PI3K pathway in lungs is beneficial in a model of airway inflammation.

**CL27c improves lung function in Th2-driven chronic asthma.** CL27c administration was assessed in a chronic therapeutic setting (Fig. 3a), and was found to significantly decrease inflammatory cells in BALF over a range of 60 to 80% (Fig. 3b). At the same time, CL27c prevented OVA-induced metaplasia of goblet cells and mucus production (Supplementary Fig. 5a, b). In addition, CL27c significantly reduced the tissue levels of eosinophil peroxidase (EPO) and myeloperoxidase (MPO) by 55 and 50%, respectively (Fig. 3c). Remarkably, prolonged treatment with CL27c achieved the anti-inflammatory effects of systemic treatment with dexamethasone (Fig. 3b, c). Similarly to corticosteroids, repeated administration of CL27c induced a lowering of Th2 mediators[21] like IL-5, IL-13, and Eotaxin (Fig. 3c, d), thus indicating that inhibition of the PI3K pathway can block Th2-mediated responses. In allergic asthma, Th2 responses are counterbalanced by T regulatory cells (Treg)[22], a cell type that significantly relies on PI3K signaling for its development and

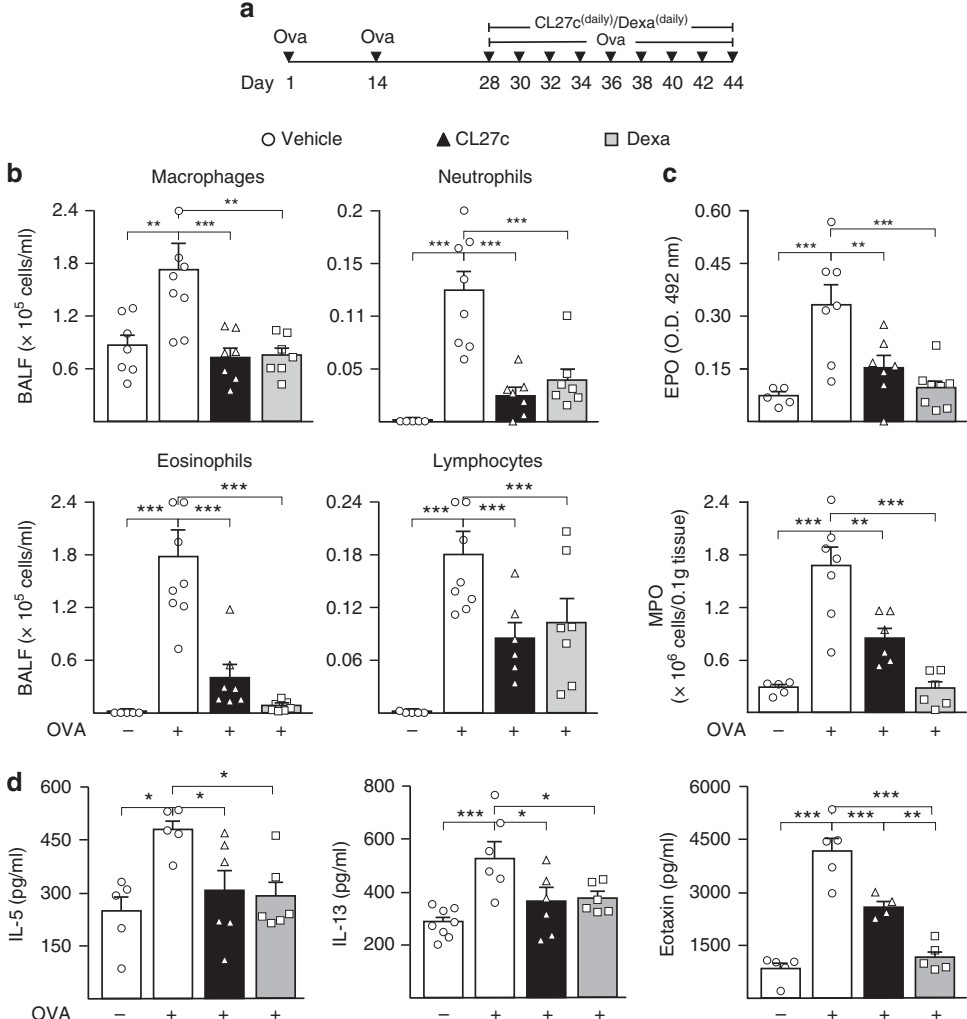

**Fig. 3** CL27c decreases leukocyte infiltration into the lungs and production of inflammatory markers in OVA-induced chronic allergic asthma model. **a** Protocol of OVA-induced chronic allergic asthma and daily treatment regimen with aerosolized CL27c (2 mg/ml) or Dexamethasone (Dexa, 5 mg/kg, i.e.; positive control). **b** Quantification of infiltrating macrophages, neutrophils, eosinophils, and lymphocytes in BALF after treatment with either CL27c or Dexamethasone (Dexa) (from left to right, macropages $n = 7, 8, 7, 7$, neutrophils $n = 5, 8, 7, 7$, eosinophils $n = 5, 8, 7, 7$, and lymphocytes $n = 5, 8, 6, 7$ independent experiments, respectively). **c** Measurement of the accumulation of neutrophil myeloperoxidase (MPO) and eosinophil peroxidase (EPO) in lung tissues after treatment with either CL27c or Dexamethasone (from left to right, EPO $n = 5, 7, 7, 8$ and MPO $n = 5, 7, 6, 5$ independent experiments, respectively). **d** Quantification of inflammatory cytokines (IL-5 and IL-13) and of the chemokine eotaxin after treatment with either CL27c or Dexamethasone (from left to right, IL-5 $n = 5, 5, 6, 6$, IL-13 $n = 8, 6, 6, 6$, Eotaxin $n = 5, 5, 4, 5$ independent experiments, respectively). Results represent mean ± s.e.m., *$P < 0.05$, **$P < 0.01$, ***$P < 0.001$ determined using one-way ANOVA followed by Bonferroni post-hoc test

function[23]. Therefore, the effect of CL27c was evaluated on in vitro differentiated Treg (Supplementary Fig. 6a, b and Supplementary Fig. 16). CL27c showed toxic effects only at very high concentrations (30 μM, Supplementary Fig. 6b), unlikely to be reached in vivo, in the OVA model. Given that Treg function is mediated by the production of anti-inflammatory cytokines like IL-10[24], this cytokine was analyzed in OVA-challenged mice. As shown in Supplementary Fig. 6c, treatment with CL27c increased

pulmonary abundance of IL-10, thus indicating that any potential effect of CL27c on resident Treg is negligible in terms of IL-10 production.

Next, airway inflammation and tissue damage were assessed by examining lung histopathology. Chronic exposure to OVA increased airway, vascular, and parenchymal inflammation, but treatment with CL27c blunted these effects, similarly to dexamethasone (Fig. 4a, b). As shown in Fig. 4b, increased accumulation of

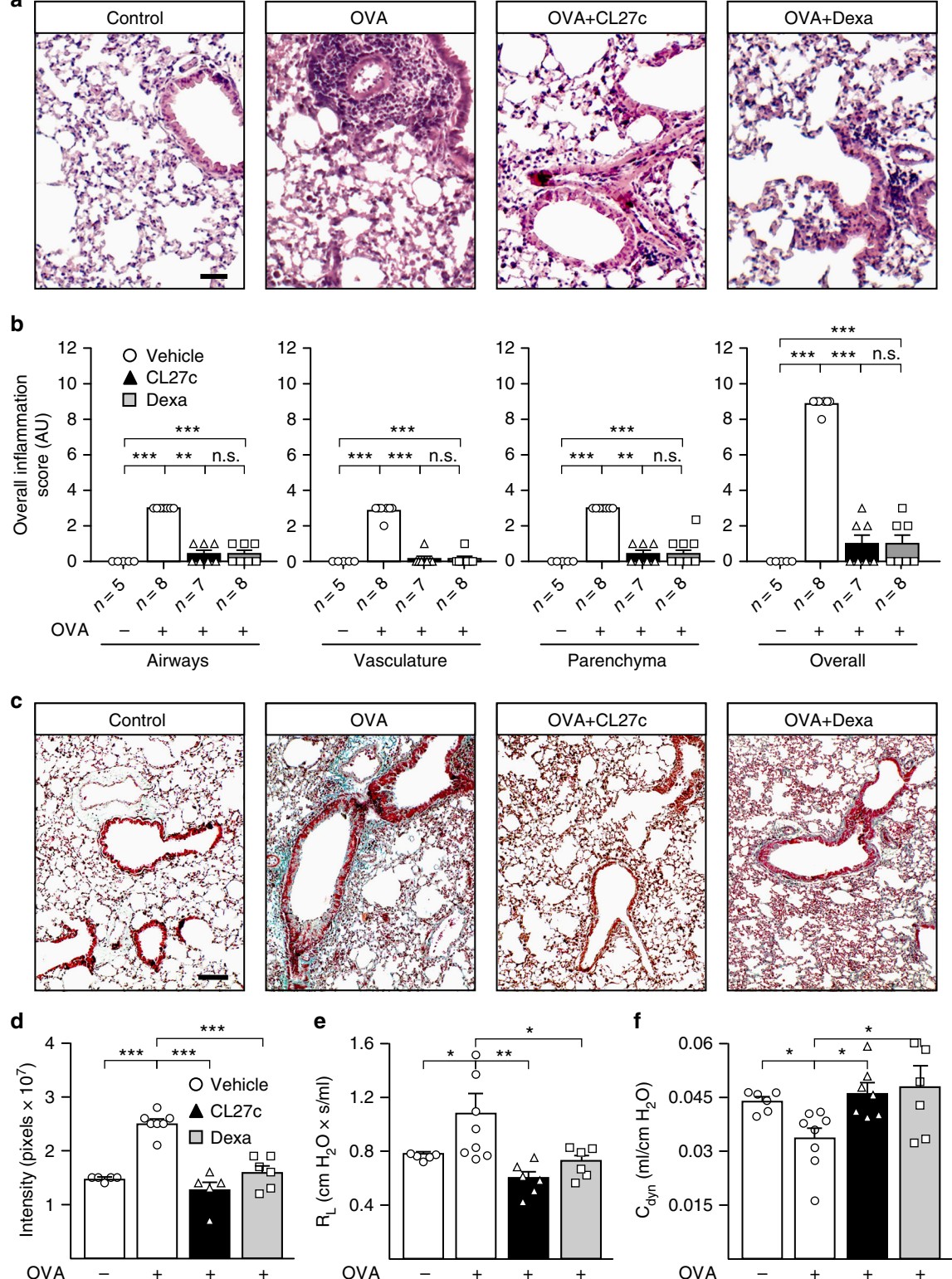

inflammatory cells in peribronchial and perivascular spaces was evident in OVA-challenged mice, but was significantly decreased in sections of lungs that received either dexamethasone or CL27c. Analysis of peribronchial fibrosis in sections stained with Gomori's trichrome revealed a pronounced fibrotic airway remodeling in the OVA group that was significantly reduced by either CL27c or dexamethasone (Fig. 4c, d). Such reduction was also supported by a reduction of transforming growth factor-β (TGF-β) and IL-13 to the level detected in unsensitized controls (Supplementary Fig. 6d and Fig. 3d).

Assessment of pulmonary function showed a markedly increased lung resistance ($R_L$) with loss of dynamic compliance ($C_{dyn}$) in the vehicle-treated group (Fig. 4e, f). However, both CL27c and dexamethasone were able to normalize lung function (Fig. 4e, f). Nonetheless, contrary to dexamethasone that, beside its anti-inflammatory action, possesses anticholinergic effects[25], CL27c was not able to modify methacholine-evoked airway hyperresponsiveness (Supplementary Fig. 7). This indicates that CL27c improves pulmonary function by reducing Th2-mediated inflammation.

**CL27c is effective in glucocorticoid-resistant asthma.** Glucocorticoids are very effective in reducing symptoms in chronic asthma and represent the mainstay of treatment. However, about 10% of the asthmatic population develops a severe form of disease marked by neutrophilic inflammation that is relatively or completely refractory to glucocorticoids[26,27]. To define whether CL27c was active in such a context, mice were let to develop a glucocorticoid-resistant neutrophilic asthma[28] and analyzed as before (Fig. 5a). OVA challenge induced an increment in airway inflammation and inhaled CL27c alone or in combination with dexamethasone was able to reduce the number of inflammatory cells in BALF (Fig. 5b), as well as MPO and EPO (Supplementary Fig. 8). Histopathological analysis revealed that, while Dexamethasone failed to provide an effect, CL27c alone was able to significantly reduce the overall score of lung damage (Fig. 5c, d). In addition, the combination of Dexamethasone and CL27c significantly attenuated tissue inflammation, indicating that PI3K inhibition is a key mechanism to restore sensitivity to glucocorticoids (Fig. 5c, d). Remarkably, this protective effect was independent of the abundance of different cytokines in BALF (Supplementary Fig. 9), thus indicating that the local action of CL27c could not block systemic production of these mediators in a model of glucocorticoid-resistant asthma. In line with its anti-inflammatory effect, treatment with CL27c alone or combined with dexamethasone restored normal lung compliance (Fig. 5e) without modifying either airway hyperreactivity or lung resistance (Supplementary Fig. 10a, b).

**CL27c attenuates bleomycin-induced pulmonary fibrosis.** PI3K signaling is not only important in asthma but also in airway remodeling and pulmonary fibrosis. Therefore, the effects of

inhaled CL27c were further studied in an in vivo model of bleomycin-elicited pulmonary fibrosis. Severe or moderate disease was induced with 3.75 or 2.00 mg/kg of bleomycin, respectively. In the severe model (Supplementary Fig. 11a), CL27c reduced infiltration of either macrophages or neutrophils and significantly decreased hydroxyproline deposition in lungs (Supplementary Fig. 11b, c) but could not protect from an observed 100% mortality within 21 days. Studies were further conducted analyzing mice in the moderate model of fibrosis, after daily exposure to aerosolized CL27c from day 10 to 21 (Fig. 6a). Treatment with CL27c markedly reduced airway inflammation (Fig. 6b) and neutrophil-derived MPO levels in the lungs (Fig. 6c). Similarly, inhaled CL27c significantly decreased the abundance of active TGF-β1 (Fig. 6d), a pro-fibrogenic factor critical for lung fibrosis initiation and perpetuation[29,30]. Accordingly, inhaled CL27c attenuated mRNA levels of *Tgfb1*, as well as of other genes involved in fibrogenesis, such as connective tissue growth factor (*Ctgf*), Collagen I (*Col1a1*), and matrix metalloproteinase-2 (*Mmp2*) (Fig. 6e).

Histopathological examination of lung tissues showed a dramatic increase of the inflammatory infiltrate at day 22 after bleomycin instillation that was reduced in CL27c-treated lungs (Fig. 7a and Supplementary Fig. 12a), representing high and low magnification, respectively), where the inflammation score appeared significantly attenuated in airways, vasculature, and parenchyma (Fig. 7b). After bleomycin instillation, a dense and diffuse interstitial lung fibrosis with loss of pulmonary architecture was observed in Gomori's and Masson's trichrome-stained lungs (Fig. 7c and Supplementary Fig. 12b, c), but this appeared dramatically reduced after CL27c treatment, which dampened fibrosis and preserved areas of lung parenchyma (Fig. 7c Supplementary Fig. 12a–c). Fibrosis was then quantified by assessing the Ashcroft score in Gomori's Trichrome-stained lung sections (Fig. 7d) as well as the collagen content in the whole organ (Fig. 7e). In both assays, inhaled CL27c was able to significantly reduce fibrotic remodeling induced by Bleomycin instillation.

In line with CL27c engaging its target and blocking the PI3K pathway in this model of pulmonary fibrosis, treatment with CL27c blunted the increase of Akt phosphorylation induced by Bleomycin administration, both in lung protein extracts and sections (Supplementary Fig. 13a, b). Given that AKT phosphorylation enhances survival of fibroblasts derived from patients with IPF[31], the effects of CL27c on AKT phosphorylation were tested in human lung fibroblasts (HLF) derived from healthy and IPF patients. In IPF cells, CL27c effectively blunted serum-induced AKT phosphorylation (Supplementary Fig. 13c). In addition, CL27c reduced survival at 1 and 10 μM in all HLF lines analyzed (Supplementary Fig. 13d), thus indicating that CL27c blocks fibrosis by directly targeting lung fibroblasts.

**CL27c lowers mortality due to pulmonary fibrosis.** To further evaluate the role of CL27c in pulmonary dysfunction induced by

**Fig. 4** CL27c maintains normal lung architecture and function in the model of OVA-induced chronic allergic asthma. **a** Hematoxylin and eosin staining of lung sections in control and OVA-treated mice further receiving either CL27c (2 mg/ml) or Dexamethasone (5 mg/kg). Bar = 25 μm. **b** Histopathological changes in lung inflammation, scored as described in Methods, in control (OVA−) and OVA-treated (OVA+) mice further receiving either CL27c or Dexamethasone (Dexa) (from left to right, airways $n = 5, 8, 7, 8$, vasculature $n = 5, 8, 7, 8$, parenchyma $n = 5, 8, 7, 8$, and overall $n = 5, 8, 7, 8$ independent experiments, respectively). **c** Representative Masson's trichrome staining showing areas of fibrosis (blue-staining) in the OVA group. Bar = 100 μm. **d** Quantification of fibrosis confirming the antifibrotic effect of either CL27c or Dexamethasone (from left to right $n = 5, 7, 5, 6$, independent experiments, respectively). **e** Analysis of Lung Resistance (RL) (form left to right $n = 5, 8, 6, 6$ independent experiments, respectively) and **f** dynamic compliance (Cdyn) (from left to right $n = n = 6, 8, 7, 6$ independent experiments, respectively) in mice with chronic asthma. Results represent mean ± s.e.m., *$P < 0.05$, **$P < 0.01$, ***$P < 0.001$ determined using either Kruskal–Wallis followed by Dunn's test (**b**) or one-way ANOVA followed by Bonferroni post-hoc test (**d-f**). n.s. = non-significant ($P > 0.05$)

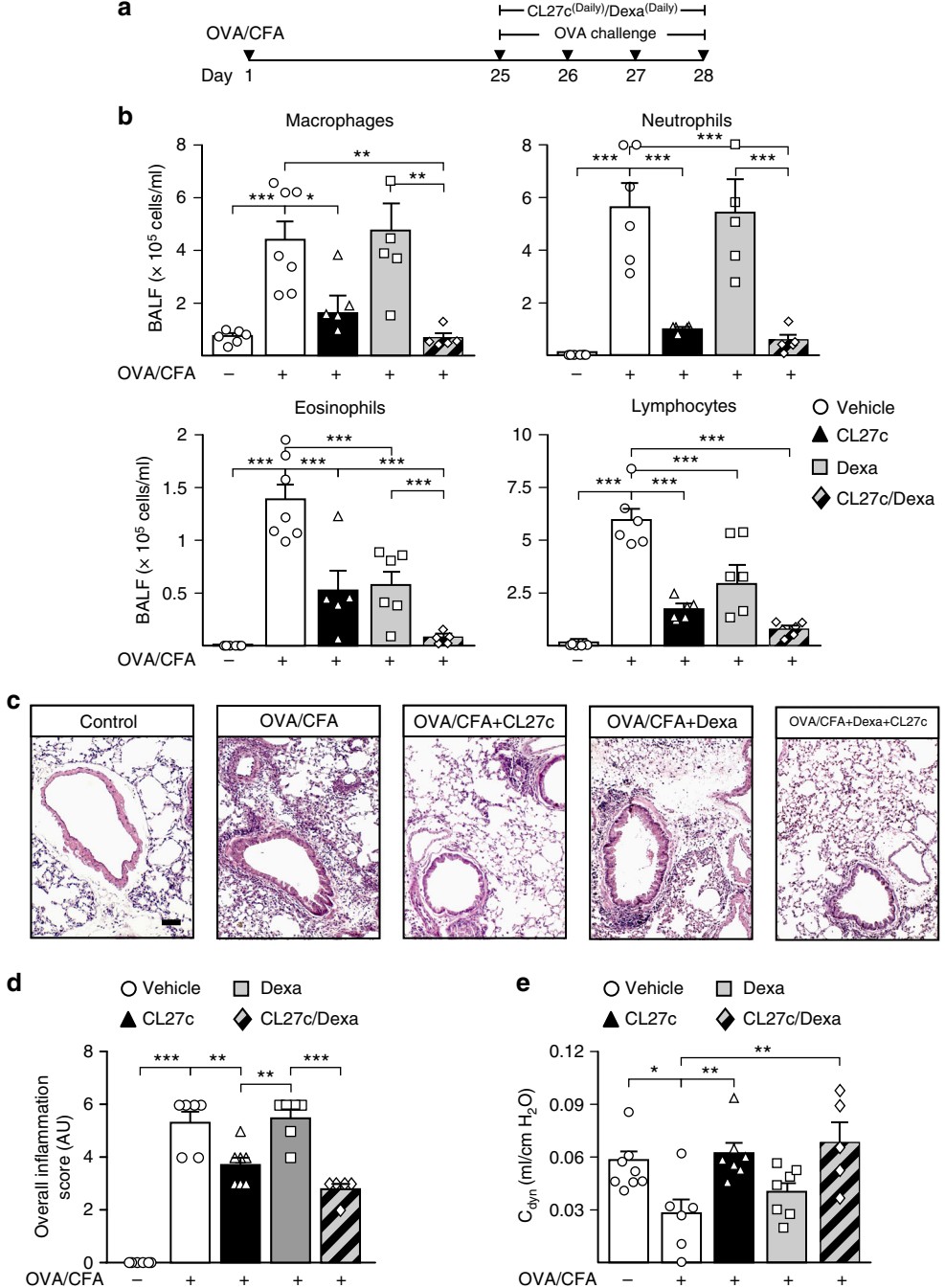

**Fig. 5** Combination of CL27c and Dexamethasone reduces inflammatory injury and prevents loss of lung function in a mouse model of corticosteroid-resistant allergic asthma. **a** Protocol for corticosteroid-resistant allergic asthma and treatment regimen with aerosolized CL27c (2 mg/ml), Dexamethasone (Dexa, 5 mg/kg i.p.), or a combination of CL27c and Dexamethasone. **b** Quantification of different inflammatory cell populations in BALF of mice treated as in (**a**) (from left to right, macrophages $n = 6, 7, 5, 5, 5$, neutrophils $n = 5, 6, 5, 5, 5$, eosinophils $n = 6, 7, 5, 6, 5$, and lymphocytes $n = 6, 6, 6, 6, 6$ independent experiments, respectively). **c** Histopathological analysis of lung sections stained with hematoxylin and eosin. Treatment with CL27c reduces the inflammatory infiltrate (black arrowheads). **d** Histopathological changes in lung inflammation, scored as described in Methods, in control (OVA/CFA−) and OVA/Complete Freund's Adjuvant-treated (OVA/CFA+) mice further receiving either CL27c, Dexamethasone (Dexa), or their combination (from left to right $n = 6, 6, 7, 6, 5$ independent experiments, respectively). **e** Analysis of dynamic pulmonary compliance (Cdyn) in control and asthmatic mice treated with either CL27c, Dexamethasone or their combination (from left to right $n = 8, 6, 7, 7, 5$ independent experiments, respectively). Results represent mean ± s.e.m., *$P < 0.05$, **$P < 0.01$, ***$P < 0.001$ determined using one-way ANOVA followed by Bonferroni post-hoc test (**b**, **c**) and Kruskal–Wallis followed by Dunn's post-hoc test (**d**)

bleomycin, lung mechanics and pulmonary alteration in elasticity were measured (Fig. 8a–g). Compared to saline/vehicle-treated mice, challenge with bleomycin resulted in a flattened and downward-shifted pressure–volume curve, corresponding to

restrictive lung disease (Fig. 8a). This resulted in a reduction of lung volume and airway flow, with a decrease of total lung capacity (TLC) (Fig. 8b), inspiratory capacity (Fig. 8c), forced vital capacity (Fig. 8d) and forced expiratory volume (FEV)

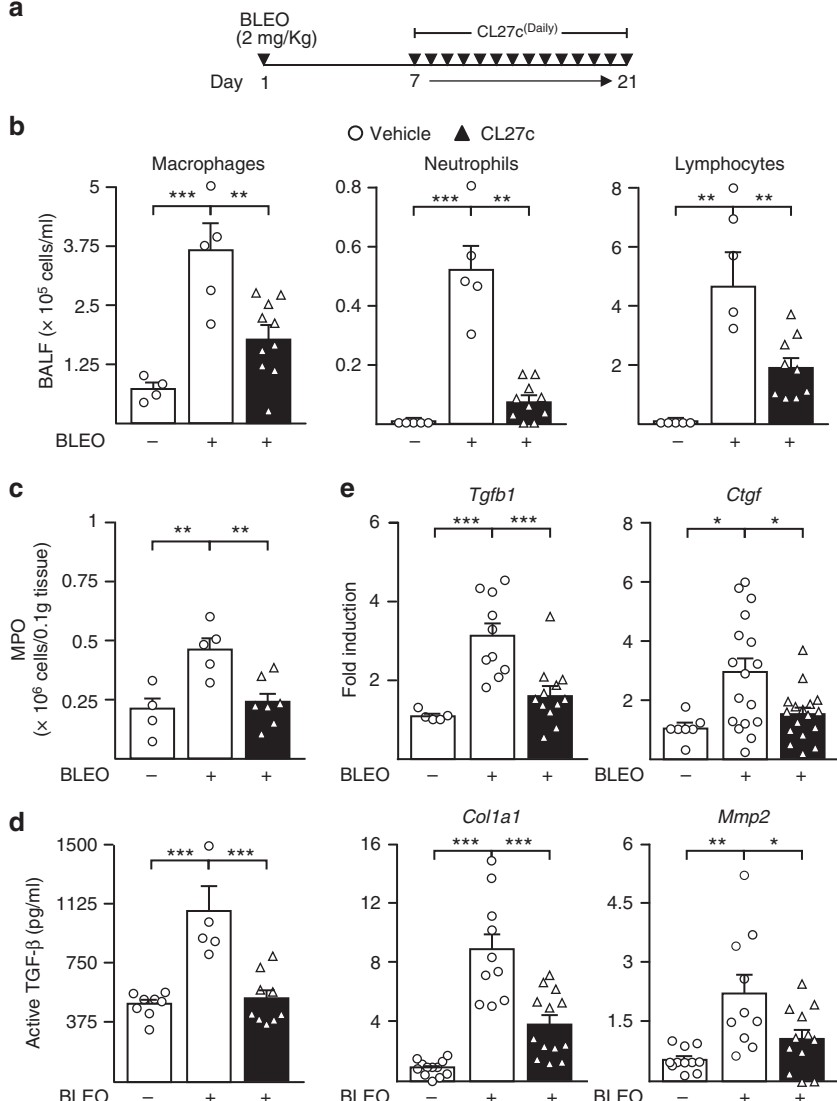

**Fig. 6** Reduced inflammation and expression of fibrotic markers after CL27c inhalation in a moderate protocol of bleomycin-induced lung fibrosis. **a** Protocol for Bleomycin-induced lung injury, elicited by intranasal instillation of 2 mg/kg Bleomycin (Bleo), and treatment regimen with aerosolized CL27c (2 mg/ml). **b** Quantification of macrophages, neutrophils, and lymphocytes content in BALF in control and CL27c-treated mice (from left to right macrophages $n = 4$, 5, 10, neutrophils $n = 5$, 5, 10, and lymphocytes $n = 5$, 5, 9 independent experiments, respectively). **c** Analysis of myeloperoxidase (MPO) expression in the lungs of control and CL27c-treated mice (from left to right $n = 4$, 5, 7 independent experiments, respectively). **d** Measurement of active levels of TGF-β with or without CL27c inhalation (from left to right $n = 8$, 5, 9 independent experiments, respectively). **e** Quantification of the expression of selected pro-fibrotic genes, including transforming growth factor-β (*Tgfb1*), connective tissue growth factor (*Ctgf*), collagen type-1 (*Col1a1*), and matrix metalloproteinase-2 (*Mmp2*) (from left to right, *Tgfb1* $n = 5$, 10, 12, *Ctgf* $n = 7$, 17, 17, *Col1a1* $n = 12$, 10, 13, and *Mmp2* $n = 11$, 10, 12 independent experiments, respectively). Results represent mean ± s.e.m., *$P < 0.05$, **$P < 0.01$, ***$P < 0.001$ determined using one-way ANOVA followed by Bonferroni post-hoc test

(Fig. 8e). Remarkably, therapeutic administration of aerosolized CL27c significantly improved the pressure–volume curve, almost restoring it to the level of control healthy mice (Fig. 8a). Consistent with this improvement, inhaled CL27c induced a significant recovery back to normal levels in TLC, inspiratory capacity, forced vital capacity, and FEV, respectively (Fig. 8b–e).

Tissue fibrosis induced by bleomycin progressively reduces lung elasticity by aberrant collagen accumulation in parenchyma, with progressive gain of lung resistance (Fig. 8f) and reduction in pulmonary compliance (Fig. 8g). On the contrary, CL27c was able to reduce the gain in lung resistance almost to the level of healthy mice and to significantly restore lung elasticity to 75% of controls. Thus, CL27c-treated mice appeared protected from the

pulmonary dysfunction induced by bleomycin. As a consequence, whereas the moderate dose of bleomycin induced 50% mortality 22 days after instillation (Fig. 8h), a 100% survival was reached after therapeutic inhalation of CL27c (Fig. 8h). Collectively, these data indicate that inhaled CL27c and consequent local blockade of the PI3K pathway in lungs reverses bleomycin-induced fibrotic disease.

**CL27c reduces persistent fibrosis**. In order to understand whether CL27c is able to improve fibrotic remodeling and is effective in reducing persistent fibrosis, inhaled CL27c treatment started at day 17 after bleomycin instillation and administered daily until

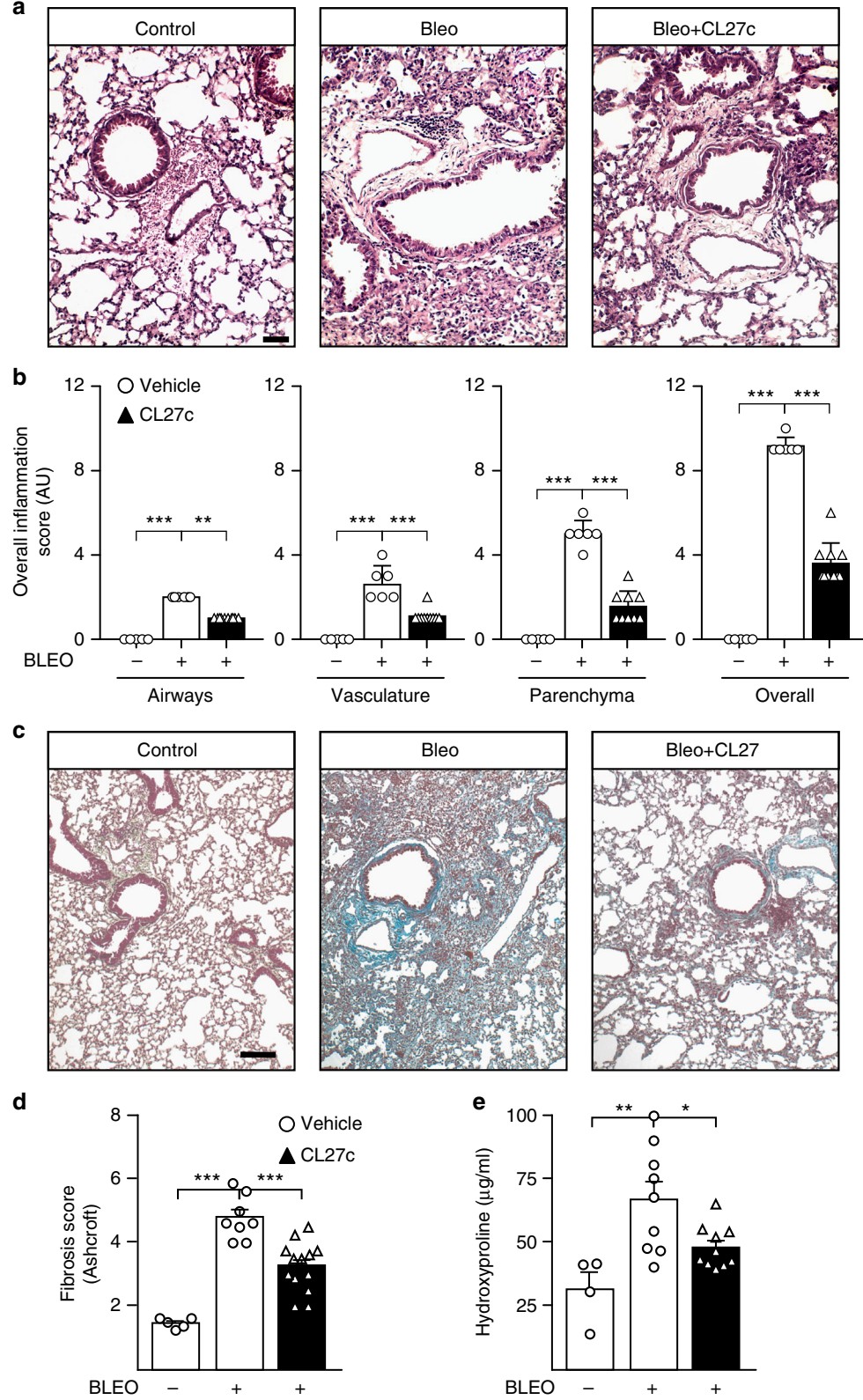

day 27. The late treatment with CL27c was still effective in reducing fibrotic collagen deposition (Supplementary Fig. 14a, b) and improved lung function and significantly blunts mortality (Supplementary Fig. 14c, d). These data showed that targeting PI3K signaling pathway has an antifibrotic effect which is independent of its anti-inflammatory activity.

## Discussion

PI3Ks inhibitors have been considered for the treatment of asthma and pulmonary fibrosis[32,33], but toxicity of systemic PI3K blockade is a major downside[7]. For example, hyperglycemia and gastrointestinal toxicity are the most striking on-target side effects of class I PI3K inhibition, often leading to treatment

**Fig. 7** CL27c attenuates inflammatory and fibrotic processes in a moderate protocol of bleomycin-induced lung fibrosis. **a** Representative hematoxylin- and eosin-stained lung tissues depicting that aerosolized CL27c (2 mg/ml) reduced Bleomycin (2 mg/kg)-elicited tissue injury and inflammatory cells accumulation. Bar = 25 μM **b** Histopathologic scoring of inflammatory damage in lung sections derived from control (Bleo−) and Bleo-treated mice (Bleo +) with and without treatment with CL27c (black and white bars, respectively) (from left to right, airways $n = 5, 5, 10$, vasculature $n = 5, 6, 10$, parenchyma $n = 5, 6, 9$, and overall $n = 5, 6, 10$ independent experiments, respectively). **c** Representative Gomori's trichrome-stained sections showing that treatment with CL27c reduced lung fibrosis, remodeling, alveolar walls, and collagen deposition (green). Bar = 100 μm. **d** Quantification of fibrosis on lung sections using the semi-quantitative Ashcroft scale (from left to right $n = 5, 8, 10$ independent experiments, respectively). **e** Analysis of hydroxyproline deposition (from left to right $n = 4, 9, 10$ independent experiments, respectively). Results represent mean ± s.e.m., *$P < 0.05$, **$P < 0.01$, ***$P < 0.001$ determined using either Kruskal–Wallis followed by Dunn's test (**b**) or one-way ANOVA followed by Bonferroni post-hoc test (**d**, **e**)

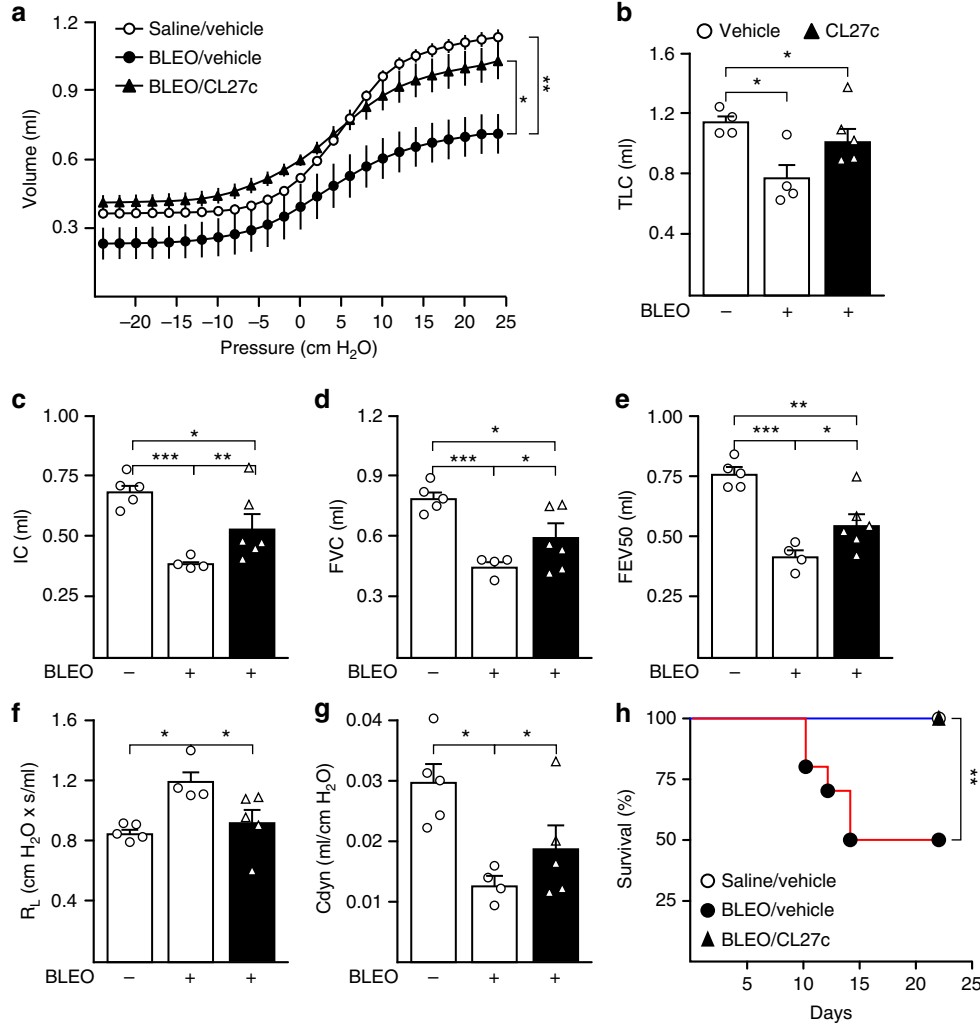

**Fig. 8** CL27c treatment restores normal lung function and improves survival in a moderate model Bleomycin-induced lung fibrosis and dysfunction. **a** Analysis of the pressure–volume curve in control mice ($n = 14$) and in mice that developed Bleomycin-induced lung dysfunction after treatment with vehicle ($n = 12$) or CL27c ($n = 12$; 2 mg/ml). **b** Analysis of total lung capacity (TLC) as in A (from left to right $n = 4, 4, 5$ independent experiments, respectively). **c** Analysis of inspiratory capacity (IC) as in (**a**) (from left to right $n = 5, 4, 6$ independent experiments, respectively). **d** Measurement of Forced vital capacity (FVC) as in (**a**) (from left to right $n = 5, 4, 6$ independent experiments, respectively). **e** Analysis of forced expiratory volume at 50 millisec (FEV50) as in (**a**) (from left to right $n = 5, 4, 5$ independent experiments, respectively). **f** Quantification of lung resistance (RL) as in (**a**) (from left to right $n = 5, 4, 5$ independent experiments, respectively). **g** Measurement of dynamic compliance (Cdyn) as in (**a**). **h** Survival curves of mice treated with either saline (white circles; $n = 8$) or Bleomycin in the presence (black triangles; $n = 10$) or absence (black circles; $n = 10$) of CL27c. Results represent mean ± s.e.m., *$P < 0.05$, **$P < 0.01$, ***$P < 0.001$ determined using one-way ANOVA followed by Bonferroni post-hoc test. Animal survival was analyzed by the Mantel–Cox log rank test

discontinuation[7]. Selective targeting of specific PI3K isoforms was thought to reduce toxicity, but emerging redundancy within the PI3K family has limited the success of such drugs[7,13,34]. Alternatively, systemic treatment with drugs targeting PI3K downstream effectors was attempted but the mTOR inhibitor everolimus was found to cause serious adverse events, including noninfectious pneumonia, and to reduce overall survival in IPF[35]. Apparently, these are off-target side effects, since pan-PI3K inhibitors, which act upstream of mTOR, do not trigger lung toxicity[36].

Differently from conventional PI3K inhibitors, CL27c was designed as a biologically inactive membrane-permeable prodrug optimized for more effective and safer inhaled delivery. CL27c is inactive but cell permeable and, once inside the cytoplasm, this compound is converted into the active form, CL27e, a hydrophilic species that cannot cross cell membranes[14]. When inhaled as an aerosol, the inactive form CL27c penetrates lung cells, where it is hydrolyzed into the active compound CL27e, thus inhibiting PI3Ks only locally, in the first encountered cells. In addition, the amount of CL27c that remains unabsorbed is converted outside cells into CL27e, a non-cell permeable, and rapidly excreted molecule with undetectable systemic pharmacological activity. Accordingly, pharmacokinetic analysis of the inhaled compound showed minimal plasma concentration and short half-life that avoided the exposure of organs other than the lung. In further agreement, when CL27c was administered per os, no significant amount of the cell-permeable compound was systemically absorbed, thus preventing side effects in case of accidental swallowing or ingestion, a major advantage for airway treatment over other currently orally available PI3K targeting compounds. In line with this view, inhaled CL27c was not able to modify either insulin signaling outside the lungs or glycemia.

Aerosolized CL27c was not only safer than classical PI3K inhibitors but also effective in controlling airway inflammation. In OVA-induced models of asthma and in bleomycin-induced lung injury, CL27c reduced Akt phosphorylation, leukocyte recruitment, and tissue remodeling in agreement with the inhibition of key PI3K-dependent innate immune responses. Although a dose response was attempted, the relative insolubility of CL27c did not allow to test doses as high as 20 mg/ml, potentially limiting the therapeutic window. Whether higher dosages can be obtained with an improved aerosol formulation has yet to be determined. Nonetheless, our findings with the potentially optimal dose of 2 mg/ml were compatible with targeting of pulmonary PI3Kδ and PI3Kγ. These enzymes are the main PI3K isoforms regulating macrophage and neutrophil trafficking in response to chemokines and chemoattractants[7] and responsible for lung accumulation of inflammatory cells in various models of airway inflammation[37].

Asthma is a primarily Th2-dominant disorder and PI3K inhibitors are known blockers of this response[38]. In agreement and in line with our findings, mice with genetic deletion of either PI3Kδ or PI3Kγ are resistant to Th2-driven asthma and show reduced levels of IL-5 and IL-13[39,40]. On the other hand, severe asthma also involves a Th17 component but whether PI3Ks play a role in Th2/Th17 polarization is controversial[41,42]. Nonetheless, glucocorticoid-resistant asthma also involves a Th1 response[43] but PI3K inhibition in respiratory diseases is known to spare interferon-γ (IFN-γ) production and Th1 responses[38]. In agreement, our findings indicate that pan-PI3K inhibition is not involved in the production of IL-17A/IFN-γ, but that PI3K signaling is critically required for the response to multiple cytokines and for the recruitment of leukocytes to the lungs.

Similarly to asthma, the model of bleomycin-induced fibrosis induces an initial inflammatory response that is rapidly substituted by massive fibroblast proliferation[44]. Our results on the efficacy of CL27c in the treatment of pulmonary fibrosis when the inflammatory reaction has already resolved as well as our studies in primary fibroblasts derived from IPF patients indicate that targeting of PI3K signaling significantly dampens fibroblast expansion and fibrotic remodeling. Inhaled CL27c attenuated mRNA levels of *Tgfb1*, as well as of other genes involved in fibrogenesis, such as *Ctgf*, *Colla1*, and *Mmp2*, thus suggesting that improvement of lung function is not only due to

the anti-inflammatory effect, but also to a modulation of processes underlying the fibrotic response. While immune responses can modulate the fibrotic responses, IPF develops independently of inflammation and drugs addressing fibroblast proliferation are currently missing or insufficiently effective. For diseases like IPF, Nintedanib and Pirfenidone can slow disease progression but none of them is effective in reducing mortality[45]. Furthermore, 20% of patients require treatment interruption due to severe side effects[45]. Activation of the PI3K pathway is currently emerging as a key pathogenic event linking inflammation and fibrosis[32,46,47]. In line with this view, PI3Kγ-deficient mice show decreased accumulation of leukocytes in the BALF and lungs, reduced lung collagen deposition, and reduced expression of pro-fibrogenic genes in a bleomycin-induced mouse model of pulmonary fibrosis[10]. In addition, mice with double mutation in ras-binding domain of PI3Kβ are resistant to bleomycin-induced lung fibrosis, which is associated with reduced activity of PI3K[11]. Simultaneous inhibition of PI3Kα/δ/β is also capable of blocking bleomycin-induced collagen production[48], thus suggesting that several PI3K isoforms orchestrate different pathogenic events that culminate into airway remodeling and fibrosis. In agreement with a major role of multiple PI3K isoforms in fibrosis, CL27c was able to reduce lung damage and prevent animal mortality, even when administered in a therapeutic setting long after the onset of the disease, at a time point where persistent fibrosis is independent of inflammation[49,50].

In summary, our observations showed that inhaled CL27c can minimize the toxic effects of systemic exposure. More importantly, our results indicating that a pan-PI3K targeting inhaled prodrug is beneficial in models of severe asthma and irreversible pulmonary fibrosis might open the way to future clinical applications. Nonetheless, such animal models have known limitations and results failed to translate to patients. Therefore, clinical testing of CL27c is the obligatory way to conclusively consolidate our preclinical results.

## Methods

**Animal models.** C57BL/6J and BALB/c mice were used according to the Italian D.L. No. 116 of 27 January 1992 and in accordance with 2010/63/EU. All study protocols were carried out accordingly to institutional guidelines in agreement with national and international law and approved by the Italian Ministry of Health with the authorization number 757/2016-PR, released on 26 July 2016 in accordance with the art.31,D.lgs.26/2016 and by the Brazilian Animal Ethics committee (Comitê de Ética em Experimentação Animal/Universidade Federal de Minas Gerais (CEUA/UFMG)) with the authorization number 82/2012 and 88/2015.

**Acute asthma model.** On Days 1 and 14, BALB/c mice (6–8 weeks) received an intraperitoneal injection (i.p.) of a solution containing 0.5 mg/ml OVA and 5 mg/ml Alum (described in Fig. 2). Untreated controls were injected with a solution of 5 mg/ml Alum. On days 25, 26, 27, and 28 the animals were challenged with intranasal injection of OVA (2 mg/ml diluted in PBS, 25 μl/mouse) and exposed for 30 min to aerosolized vehicle (0.5% tyloxapol/saline solution), CL27c (2 mg/ml dissolved in 0.5% tylaxopol/saline solution) or to systemic treatment with dexamethasone (5 mg/kg i.p. injection). Untreated control groups included mice challenged with PBS alone. On day 29, mice were sacrificed, and samples were collected for further analysis.

**Analysis of lung delivery.** Mice were subjected to a 30-min aerosolized treatment with CL27c or vehicle (0.5% tyloxapol/saline solution)-containing solutions, where tyloxapol was added to improve distribution in the lungs. Mice were anesthetized at 1, 3, and 6 h after aerosolized treatment, the trachea was exposed and a 1.7-mm-outside-diameter polyethylene catheter was inserted and they were subsequently sacrificed. A syringe with a needle (27 G) pretreated with heparin was inserted into the vena cava above the renal veins to collect blood. After the collection of blood, lungs were subjected to bronchoalveolar lavage with three different 500 μl aliquots of acetonitrile. At the same time blood samples were centrifuged 180× *g* at 22 °C for 10 min, plasma supernatant was collected and treated with 1 volume of acetonitrile to precipitate proteins. Samples were then agitated on a high-speed vortex for 2 min and further mixed

using a rotary shaker for 2 min. the precipitated proteins were then pelleted by centrifugation at $10,000 \times g$ for 10 min and the supernatant was transferred in a new tube. BALF and plasma extracts were subsequently subjected to Liquid chromatography-mass spectrometry.

A Nexera LC-30AD (Shimadzu, Milan, Italy) instrument equipped with a Luna C18 $150 \times 2.1$ mm, 3 µm particle size column (Phenomenex, Milan, Italy) was used to carry out chromatography analysis. The eluents (VWR, Milan, Italy) were acetonitrile (A) and 0.1% formic acid (B) in the following gradient conditions: from 10 to 54% of solvent A in 21 min, up to 100% of A in 2 min and re-equilibration. The injection volume was 10 µL, the flow rate was 200 µL/mL and the column was maintained at the temperature of 40 °C. For the chromatographic separation, an octadecylsilica column was used. The employed mass analyzer was a Q-trap used in multiple reaction-monitoring (MRM) mode. With this technique, it was possible to detect and quantify simultaneously the drug and the metabolite with high selectivity and sensitivity. A QTrap-5500 (Sciex, Milan, Italy) instrument, dressed with a Turbo Ion Spray source, was utilized to analyze samples. The source parameters were: curtain gas 30 (arbitrary units, a.u.), gas1 40 (a.u.), gas2 55 (a.u.), temperature 500 °C, ion spray voltage 5500 V, declustering potential 200 (a.u) and entrance potential 11 (a.u.). The detector was used in MRM mode and the transitions for each compound were: $446 > 418$ m/z and $446 > 401$ m/z for CL27C (collision energy 29, collision exit potential 26 and 28, respectively); $432 > 387$ m/z and $432 > 404$ m/z for CL27e (collision energy 30 and 28, collision exit potential 26 and 27, respectively).

**Analysis of p-Akt in CLL cells, neutrophils, and BMDMs.** CLL cells (#Hs 505.T, ATCC® CRL-7306™) were cultured with RPMI additioned with 10% FBS and 100 µg/ml penicillin and streptomycin. The day of the assay CLL cells were seeded in 96-well plate at a concentration of $7 \times 10^4$ cells/ml in starving medium, RPMI free. The cells were starved for 6 h and subsequently pretreated for 1.5 h with CL27c at different concentrations. The cells were stimulated for 15 min with 15 µg/ml of goat anti-Human IgM-MU chain specific antibody (Sigma-Aldrich #I0759-1MG) and proteins were extracted with Laemmli buffer.

BMDMs were obtained from bone marrows isolated from femurs and tibias of C57BL/6J mice. Bone marrow cells were cultured in a petri plate in the presence of RPMI additioned with 30% L-cell conditioned medium (supernatant of 5 days culture of L929 cells), 20% FBS and 100 µg/ml penicillin and streptomycin. After about 7 days, almost all cells became attached BMDM. BMDM were pretreated with Versene (GIBCO™ #15040066) to detach them from the plate, washed, and placed on a 24 wells culture plate at a density of $2 \times 10^5$ cell/well in RPMI free medium. After 6 h of starvation, BMDMs were pretreated with CL27c at different concentrations and stimulated with C5a (25 nM, Sigma-Aldrich #SRP4895A) for 5 min. Proteins were extracted with Laemnli buffer.

Neutrophils were obtained from bone marrow isolated from femurs and tibias of C57BL/6J mice. Isolated bone marrow cells were centrifuged 12 min at $260 \times g$ and resuspended in 2 ml of physiologic salt solution and neutrophils were subsequently isolated after centrifugation (30 min at $1100 \times g$, 4 °C without brake) through 3 Percoll gradients (from bottom to top: solution B, 75% of a 90% Percoll and 10% HBSS stock solution; solution C, 64% of a 90% Percoll and 10% HBSS stock solution; solution D 52% of a 90% Percoll and 10% HBSS stock solution). Once collected at the interface between the solution B and solution C, the cells were washed three times with 10 ml of physiologic salt solution and centrifuged for 5 min at $260 \times g$, 4 °C and pretreated for 1 h with CL27c at different concentration. After stimulation with C5a protein were extracted with Laemnli buffer.

For all cell types, phosphorylated Akt (P-Akt, Ser 473, 1:100, Cell Signaling Technology #4060) production was detected by western blot analysis and quantified with ChemiDoc™ XRS system. P-Akt signals are normalized on endogenous. For the calculation of EC50 values, a percentage of residual P-Akt was calculated for each inhibitor concentration by using the control vehicle as 100%. To derive the EC50, all data were plotted on a dose response curve (Graph Pad software) and the EC50 was calculated by using the non-linear regression fit (equation [log agonist] vs response–variable slope).

**Assessment of leukocytes in airways and lungs.** BALF was performed to obtain leukocytes from alveolar space. The trachea was exposed and a 1.7-mm-outside-diameter polyethylene catheter was inserted. BAL was performed by washing the lungs three-times with three different 500 µl aliquots of PBS. BALF samples were centrifuged at $600 \times g$ for 10 min at 4 °C on a histology glass slide and stained with Diff Quick (MICROPTIC, #DQ-ST). The number of leukocytes and differential counts of different population were obtained evaluating the percentage leukocyte populations over the total cell number.

**Assessment of respiratory mechanics.** Mice were anesthetized with a subcutaneous injection of ketamine and xylazine (130 mg/Kg ketamine and 8.5 mg/Kg xylazine) to maintain spontaneous breathing under anesthesia. Mice were tracheostomized, placed in a body plethysmograph and connected to a computer-controlled ventilator (Forced Pulmonary Maneuver System®, Buxco Research Systems©, Wilmington, North Carolina USA). First, an average breathing frequency of 160 breaths/min was imposed to the anesthetized animal by a pressure-

controlled ventilation until a regular breathing pattern and complete expiration at each breathing cycle was obtained. Under mechanical respiration, respiratory flow and airway pressure were measured to compute dynamic compliance (Cdyn) and lung resistance ($R_L$). To measure the TLC, pressure–volume maneuvers were performed, where lungs were inflated to a standard pressure of $+30$ cm $H_2O$ and then let slowly exhale until a negative pressure of $-30$ cm $H_2O$ was reached. To measure the FEV Fast-Flow Volume maneuvers was performed. In this maneuver, lungs were first inflated to $+30$ cm $H_2O$ (TLC) and, immediately, a negative pressure ($-30$ cm $H_2O$) was applied in order to enforce expiration. To evaluate AHR, the same mice used in previous maneuvers (basal condition) received Metacholine, 1 mg/Kg (Acetyl-β-methylcholine chloride, A-2251, Sigma-Aldrich St.Louis, MO,USA) i.v., and 20 sec after, a new set of maneuvers were conducted to assess $R_L$ changes. Suboptimal maneuvers were rejected and for each test in every single mouse at least three acceptable maneuvers were conducted to obtain a reliable mean for all numeric parameters.

**Bleomycin-induced lung injury and fibrosis in mice.** Eight-to-ten-week-old C57BL/6J males were used in bleomycin-induced lung injury and fibrosis model[10,51]. Briefly, a single 40 µl injection containing 3.75 or 2 mg/kg of bleomycin sulfate (Sigma Aldrich-MERCK #B1141000) diluted in PBS or PBS only (vehicle) was instilled intratracheally. From day 7 to 21 mice were treated daily with CL27c (2 mg/ml dissolved in 0.5% tylaxopol/saline solution, aerosolized for 30 min) or vehicle (0.5% tyloxapol/saline solution, aerosolized for 30 min). On day 22, mice were sacrificed, and samples were collected for further analysis.

**BMDM migration.** For the migration assay, BMDMs (isolated as above described) were pretreated with Versene (Gibco™ #15040066) to detach them from the plate, washed, and incubated with free RPMI with different concentrations of CL27c for at least 2 h. Migration assay was subsequently performed by using a Boyden chamber toward C5a (50 nM, Sigma-Aldrich #SRP4895A) and CXCL12 (100 nM, Peprotech #250-20A) chemokines for 1 h at 37 °C, 5% $CO_2$. Migrated BMDMs, attached to the polycarbonate of the Boyden chamber, were stained with Diff-Quik® reagent (MICROPTIC S.L. #DQ-ST) and counted at the microscope (5 fields/sample). The percentage of migrated cells was calculated by relating the number of the treated cells with the control vehicle (free RPMI).

**Chronic asthma model.** On day 1, BALB/c mice received systemic immunization by subcutaneous injection of 10 µg of chicken egg ovalbumin (OVA grade V, >98% pure; Sigma, St Louis, MO) diluted in 2 mg/mL Al(OH)3 followed by a booster injection at the day 14 (described in fig. 3). Nasal challenges were performed starting at the day 28, by inhalational exposure to aerosolised ovalbumin (1% ovalbumin in saline 0,9%) for 20 min per day, in alternated days during 16 days (day 28, 30, 32, 34, 36, 38, 40, 42 and 44). At the same time, mice received daily treatment from day 28 to 44 with CL27c (2 mg/ml dissolved in 0.5% tylaxopol/saline solution, aerosolized for 30′) or vehicle (0.5% tyloxapol/saline solution, aerosolized for 30′), or dexamethasone (5 mg/kg i.p. injection). Untreated control groups included mice challenged with PBS alone. On day 45, mice were sacrificed, and samples were collected for further analysis.

**Cytokine, chemokine, and IgE analysis.** IL-5, IL-13, eotaxin, TGF-β, KC, IL-17A, and MCP-1 levels in the BALF and anti-OVA IgE levels in serum were measured by ELISA technique using commercial DuoSet kits R&D Systems according to the instructions of the manufacturer. Results were expressed in levels of cytokines per ml of lung lysate (pg/ml) or arbitrary units (A.U.).

**Glucocorticoid-resistant asthma model.** On day 1 BALB/c mice received systemic immunization by subcutaneous injection of 20 µg of chicken egg ovalbumin (OVA grade V, >98% pure; Sigma, St Louis, MO) in saline, emulsified in 75 µL CFA (Complete Freund's Adjuvant; Sigma-Aldrich) (Figure 5)[52]. On days 25–28 mice were exposed to aerosolised ovalbumin (1% in saline) for 20 min per day. Mice were exposed during 30 min to aerosolized CL27c (2 mg/ml dissolved in 0.5% tylaxopol/saline solution) or vehicle (0.5% tyloxapol/saline solution), or to systemic treatment with dexamethasone (5 mg/kg i.p. injection). Untreated control groups included mice challenged with PBS alone. On day 29, mice were sacrificed, and samples were collected for further analysis.

**Hydroxyproline assay.** Levels of hydroxyproline in lung tissues were measured using the Hydroxyproline Assay kit (Sigma-Audrich, St. Louis, MO, USA) according to the manufacturer's instructions. Briefly, 10 mg of pulverized frozen lungs homogenized in 100 µL of concentrated hydrochloric acid (HCl, 12 M), and hydrolyzed at 120 °C for 3 h. Then, 5 µL of each sample in triplicate were used to the assay. To avoid false results due substances that may interfere with the reaction, we did the spiked samples as suggest by the manufacturer, which we can quantify the interference of the sample in a standard concentration. The absorbance was obtained at 560 nm. The final results are expressed as concentration of hydroxyproline in µg/mL.

**Immunohistochemical reaction to CD18-positive cells**. Immunohistochemistry for CD18 was performed using the streptavidin-biotin-peroxidase method in formalin-fixed, paraffin-embedded tissue sections (5 μm thick) mounted on poly-L-lysine-coated microscope slides. Lung cross-sections were deparaffinized and rehydrated through xylene and graded alcohols. After antigen retrieval, endogenous peroxidase was blocked (15 min) with 3% (v/v) hydrogen peroxide and washed in phosphate-buffered saline (PBS). Sections were incubated overnight (4 °C) with primary rabbit anti-CD18 antibody (Acris Antibodies GmbH, Germany, Cat. Number B0842-1) diluted 1:400 in PBS plus bovine serum albumin (PBS-BSA) or rabbit anti/PAkt (S473) diluted 1:100 in PBS/BSA. The slides were then incubated with biotinylated goat anti-rabbit antibody (Dako Agilent Pathology Solution, USA) diluted 1:1000 in PBS/BSA. After washing, the slides were incubated with avidin-biotin-horseradish peroxidase conjugate (Dako Agilent Pathology Solution, USA) for 30 min, according to the manufacturer's instructions. CD18 and PAkt-positive cells were visualized with the chromogen 3,3′-diaminobenzidine (DAKO liquid DAB + substrate chromogen system, CA, USA). Slides were counterstained with Harry's hematoxylin, dehydrated in a graded alcohol series, cleared in xylene, and cover slipped.

**Insulin stimulation analysis in liver, heart, and lung**. Animals were fasted overnight and, the day after, they were treated for 30 min with aerosolized CL27c (2 mg/ml in 0.5 % tylaxopol/saline solution). After 1 h from the CL27c treatment mice were administrated with insulin (1.0 UI kg$^{-1}$ diluted in PBS solution, Actrapid, Novo Nordisk, i.p. injection). Ten min after injection, the animals where sacrificed and tissues removed, frozen in liquid nitrogen, and homogenized in lysis buffer (50 mM Tris-HCl, pH = 8, 150 mM NaCl, 1% Triton X-100) supplemented with 50 mM NaF, 2 mM sodium orthovanadate, 1 mM sodium pyro-phosphate, and protease inhibitor cocktail (Roche).

**Lung histopathology**. The left lung was removed and fixed in 4% neutral phosphate-buffered formalin (pH 7.4)[10]. The tissues were dehydrated gradually in ethanol, embedded in paraffin, cut into 4 μm sections, stained with H&E, Gomori's trichrome and periodic acid schiff (PAS), and examined under light microscopy by a pathologist in a double blinded manner. Images of lung sections were captured with a digital camera (Optronics DEI-470) connected to a microscope (Olympus IX70) with a magnification of ×40 or ×100. Inflammation was scored according to modification of existing protocols[53] and to the histopathology criteria reported in Supplementary Table 4.

**MPO assay**. Lung samples were homogenized in hexadecyltrimethyl-ammonium bromide buffer (50 mg of tissue/mL). The homogenates were then centrifuged at 2000 × g for 15 min at 4 °C. MPO activity in the resuspended pellets was assayed by measuring the change in absorbance at 450 nm using a reading solution (5 mg of O-dianisidine, 15 μL of 1% $H_2O_2$, 3 mL of phosphate buffer, and 27 mL of $H_2O$). The change in absorbance was recorded and plotted on a standard curve of neutrophil density. The obtained data were expressed as MPO activity (neutrophils/0.1 g of tissue).

**PBMCs proliferation**. PBMCs in RPMI medium were seeded in a 96-well tissue culture plates at $5 \times 10^3$ cells/well (Cell Star, Frickenhausen, Germany). PBMCs were activated with vehicle (PBS) or anti-CD3 (5 μg/mL) plus soluble anti-CD28 (2 μg/mL) mAbs (BD Pharmingen, San Jose, CA, USA). Then, the cells were incubated at 37 °C and 5% $CO_2$ for 48 h. Cell proliferation was assayed by MTT (3-(4,5-dimethylthiazol-2-yl)-2,5-diphenyltetrazolium bromide) assay (Roche Life Science, Mannheim, BW, Germany) following the manufacturer's instructions. Briefly, 10 μL of MTT (3 mg/mL) was added to each well containing cells in 100 μL medium and incubated for 2 h at 37 °C. Then, 100 μL of lysis buffer was added allowing the dissolution of formazan crystals. After 15 min at room temperature the absorbance was read at 560 nm with reference of 690 nm using GloMax®-Multi Microplate Multimode Reader (Promega, Madison, Wisconsin, EUA).

**Quantification of EPO in lung**. EPO activity in lung samples was determined as means to detect eosinophil recruitment. The substrate solution consisted of 0.1 mM o-phenylenediamine in 0.05 M Tris-HCl buffer pH 8.0 containing 0.1% Triton and 1 mM hydrogen peroxide. Absorbance was determined in a plate reader at 492 nm. Values are expressed in O.D.

**Real-Time PCR**. For quantitative measurements, total RNA was extracted from lung tissues using the High Pure RNA isolation kit (Invitrogen, Life Technologies, Paisley, UK) and reverse-transcribed with the Superscript first-strand synthesis system kit (Invitrogen, Life Technologies, Paisley, UK). Specific mRNA sequences were quantified by real-time PCR with primer/probe combinations based on the Roche Universal ProbeLibrary Set, Mouse (Tgfb1 probe: #72, Ctgf probe: #71, Col1a1 probe: #15 and Mmp2 probe #49). Primer sequences are listed in Supplementary Table 5.

**ROS production analysis**. Production of ROS was determined by primary isolated neutrophils ($5 \times 10^6$/mL) (as described above) with lipopolysaccharide (1 μg/mL) for 1 h and then preincubating them with luminol (20 μg/mL) and horseradish peroxidase (35 μg/mL) in RPMI medium plus 10% FCS. The cells were subsequently stimulated with 2.5 μM formyl-methionyl-leucyl-phenylalanine (fMLP, Sigma-Aldrich), and light emission was recorded every 5 sec for a period of 300 sec in a Berthold MicroLumat Plus luminometer (Berthold Technologies).

**Treg isolation and viability assessment**. Cells were obtained from Lymph nodes and spleen from naive C57BL/6. Cells were then purified with CD4 T cell isolation Kit and with CD25 biotin and microbeads anti-biotin. Purified cells were cultured for 72 h in a 96-wells plate with anti-CD3/CD28 coated (1 ug/ml), TGF-B (1 ng/ml), or Cl27C (1, 3, 10 or 30 uM). After 72 h, the cells were harvested and stained with PBS (2% Bovine serum fetal) and anti-CD4 and fixable viability dye for 10 min. Cells were then fixed and permeabilized with intracellular fixation and permeabilization buffer set (eBioscience). Anti-FoxP3 was used for permeabilization. After the staining, the cells were acquired with FACS VERSE machine and analyzed by FlowJo X software.

**Western blot analysis**. Lung, liver, and heart samples were collected for western blot detection of akt phosphorilation levels. Protein samples were resolved by 10% SDS-PAGE and transferred on to PVDF (Immobilon™-P) membranes. The membranes were blocked with 5% BSA in 1x TBS with 0.3% Tween 20 for 30 min at 42 °C. Then, the membranes were incubated overnight at 4 °C with primary antibodies (phosphorylated Akt serine 473, 1:1000, Cell Signaling Technology, Cat. Number 4060) and 1 h with HRP-conjugated secondary antibody (1:10000). The blots were visualized in ECL solution for 1 min and images were captured on Luminescent Image Analyser (ChemiDoc Imaging Systems, Bio-Rad Laboratories, USA). For loading control, the blots were striped and reassessed with pan Akt (1:1000) antibody. For quantification, specific phosphorylation was calculated as the ratio of signals for phosphorylated Akt to the signal of total Akt detected. Values of unstimulated samples were set to 1.0. Uncropped scans of the most important blots are provided as a supplementary figure in the Supplementary Information (Supplementary Fig. 15).

**Statistical analysis**. One-way or two-way ANOVA followed by Bonferroni post-hoc test was used for parametric data, which were expressed as the mean ± standard error of the mean (S.E.M.). The non-parametric data were analyzed by Kruskal–Wallis followed by Dunn's test. The Mantel–Cox log rank test was used for survival curves. Statistical significance was accepted when $P < 0.05$ and indicated as: $*p < 0.05$, $**p < 0.01$, and $***p < 0.001$.

## Data availability

The data that support the findings of this study are available within the article and Supplementary Files or available from the corresponding author upon reasonable request. A reporting summary for this Article is available as a Supplementary Information file.

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

## Acknowledgements

The authors wish to thank the following colleagues for their contribution to the pharmacokinetic studies: Katia Rolfo, Cristiano Musso, Cristiano Marozin, Enrico Vigna, Mario Marubini and Lorena Corradin (RBM-Merck Serono), and Stefano Porzio (LIMA-BIPCA). The authors also acknowledge Nanosyn Biology Inc. for performing the kinome assay. We are grateful to Dr. Fabricio Marcus Silva Oliveira and Camila Simões Freitas (Laboratory of Pulmonary Immunology and Mechanics, UFMG) as well as to Dr. Paola Circosta (Department of Clinical and Biological Sciences, University of Torino) for technical assistance. This work was supported by "Futuro e Ricerca 2010" (RBFR10HP97_004); "Fondo per lo Sviluppo e la Coesione 2007–2013–Regione Piemonte" (A11_2015_12C), DRUIDI; Regione Piemonte "Bando regionale a sostegno di progetti di ricerca industriale e/o sviluppo sperimentale sulle malattie autoimmuni e allergiche" (PARFSC 2007/2013; TIPSO project); AIRC 16813; Compagnia di SanPaolo (CSTO161109); Conselho Nacional de Desenvolvimento Científico e Tecnológico–CNPq/Brazil (Processo 474887/2011-1–Edital Universal 14/2011); Fundação de Amparo a Pesquisa do Estado de Minas Gerais–FAPEMIG/Brazil (CBB APQ-03570-16–Edital Universal 01/2016); Pro-Reitoria de Pesquisa da Universidade Federal de Minas Gerais, Brazil. R.L.S. received a Capes Fellowship (Pesquisador Visitante Especial, processo nº 88887.115812/2016-00). C.C.C. was supported by a FIRC fellowship. R.C.P.L-J was supported by a Capes Fellowship (Estágio Sênior no Exterior–Processo nº 88881.119732/2016-01) and by PRONEX/FUNCAP/CNPq (PR2-0101-00054.01.00/15).

## Author contributions

All authors contributed extensively to the work presented in this paper. C.C.C., R.L.S., J.P.M., M.S.M., and L.R.K. performed most of the experiments and developed the methodology. T.P. and G.C.T. provided the compounds, CL27c and CL27e for in vitro and in vivo experiments. M.S.M., L.R.K., D.C.R., G.D.C., and E.M.D. performed histopathology cytokines measurement and sample assays. M.S.M., L.R.K., and R.C.R. performed mice pulmonary mechanics. F.C. performed cell-based assays to test the compound. PK studies were designed and conducted by G.G., S.A., F.D.B., and C.M. Experiments on Treg cells were performed and supervised by D.P. and J.C.A.-F. E.C., R.C.R., E.H., and M.M.T. jointly conceived the original idea; E.C. and R.C.R. supervised the project, collected the data, and performed the analysis. E.H., E.C., R.C.P.L-J., V.S., and R.C.R. wrote the manuscript. All authors provided final approval of the version submitted for publication.

**Additional information**

**Competing interests:** E.H. and G.C.T. are co-founders of Kither Biotech, a company involved in the development of PI3K inhibitors. The other authors declare no competing interests.

