## [Peer Review File · Nature Communications]

Reviewer #1 (Remarks to the Author):

The studies by Campa and collaborators analyze the pharmacokinetics and efficacy of C27c, an inhaled pan-PI3K inhibitor, in preclinical models of asthma and pulmonary fibrosis. Pan-PI3kinase inhibitors have been approved for some indications; however there is toxicity with systemic and chronic administration. C27c has potential advantages to diminish toxicity since it is a cell permeable pro-drug and can be administered by inhalation. Since previous studies have shown activation of the PI3K pathway in chronic lung diseases such as asthma and fibrosis, the studies are relevant to demonstrate the potential efficacy of this new compound. Despite these translational advantages there are some concerns that the authors need to address.

Major concerns.

1. How is PI3K/Akt activation in animal models treated with C27c? Total levels and IHC analysis can also be useful to define cells with effective inhibition of this pathway during disease.
2. PI3K inhibition has been related to immune toxicity by reduction of T regulatory cells. Since T regs can have a role in airway remodeling and they have been associated with proliferation and Th2 cytokine release in response to allergens, it would be important to define if C27c causes changes in T regs populations in the asthma lung models and if cytokines like TGF- β and IL-10 levels were modified by the compound.
3. C27c studies in the ova sensitization models show significant diminution of Th2 mediated responses. However, a key role of Th1 responses have been also recognized in the pathobiology of asthma, particularly in severe types of asthma that is not responsive to corticosteroids so levels of key cytokines as IFN- γ is important to define in these models. In addition, C27c was not able to control AHR and IFN- γ has been found to be critical to control airway hyperresponsiveness in models of severe asthma (Raundhal et al JCI 2015).
4. The authors in the discussion have an erroneous interpretation of the pathogenesis of IPF "Development of pulmonary fibrosis is a consequence of tissue accumulation of activated effector immune cells in the lung". The most accepted theory of the pathogenesis of IPF is the injury of epithelial cells with activation of profibrotic responses that causes disrepair and accumulation of fibroblasts. Although immune responses can be important to modulate fibrotic responses in IPF, the main pathogenic mechanism is not the accumulation of immune cells.

On contrary, immune responses are important for the development of bleomycin induced lung fibrosis, a model that has several limitations that prevent the direct application to human IPF, as

hundreds of studies have shown. Probably, the most relevant model at this point is to use aging mice treated with bleomycin (Jenkins et al AJRCMB 2017, Tashiro et al Front Med 2017).

5. Akt activation has been found to be important for the resistance to apoptosis of lung fibroblasts derived from IPF lungs (Kulasekaran et al AJRCMB 2009). In vitro studies comparing this type of response in lung fibroblasts from donor controls and IPF lungs or the use of human lung slices will bring relevance to the translational potential of this drug. Additional studies using the drug between day 14-21 post bleomycin can also show that the drug is effective not only to control immune responses but also the persistence of fibrosis that is a key component of the disrepair in IPF.

Reviewer #2 (Remarks to the Author):

Campa et al have studied the PK/PD characteristics of a novel inhaled pro-drug (CL27) that inhibits Class 1 PI3Ks, and then demonstrated efficacy of this compound in murine models of OVA-induced acute, chronic and steroid-resistant asthma and bleomycin-induced fibrotic lung disease. The benefits of PI3K inhibition in these models are not novel; the novelty of the manuscript lies in method of active drug delivery, which holds great promise for future therapeutic intervention with restricted toxicity. Whilst I believe this is an exciting and important advance that would be of interest to the journal readership there are some issues that need to be addressed. Overall the statistical analysis seems valid, with a couple of requests for clarification given below.

Major Issues

1. My main concern is that target engagement has not been demonstrated in the relevant models, only following ip-insulin treatment (which is not a pro-inflammatory stimulus). The authors need to show data indicating the impact of CL27c on lung and systemic targets on PI3K activation in at least one of the asthma models and in the bleomycin model.
2. Figure 1A suggests inhibition of Pi3K activity to baseline at 0.3µm CL27c in 'CLL cells', yet full functional inhibition in neutrophils and PBMCs seems to required 10µm. To better relate to the

functional data, and also to the models used subsequently, the authors should demonstrate the concentration-dependence of inhibition of AKT phosphorylation in more relevant target cells (eg neutrophils and PBMCs) and should quantify effects by densitometry.

3. Suppl Fig S2/Table S4. As far as I can see, the inhibitory profile of CL27 is only given for PI3K α , the isoform likely least likely to be relevant to the models employed. Data for all other Class 1 and also Class 2/ 3 PI3Ks, plus AKT and mTOR should be included in these tables. If this is not possible for some reason, perhaps the IC50s of CL27c for all of these kinases could be presented in a separate table?

4. The figure legends are not well written – there are several typos and omissions (see below). In some cases the authors just recapitulate a summary statement of the results without indicating how the experiment was undertaken, and the latter needs to be conveyed in brief. Please could the authors re-read the legends carefully and optimise them.

5. Importantly related to the above please clarify the doses of CL27c used in each model. In the main text and the methods section it states 2mg/ml/mouse was used in all cases, and this dose is quoted in the legend for Figure 2, but in the legends for Figures 3, 4, 5 and 7 it says 0.2mg/ml/mouse was used, whilst in Figure 6 the dose is given as '2mg/kg'. It is very important that the manuscript is internally consistent on this issue.

Minor Issues and Typos

1. It should be clearly stated in the Introduction that PI3K inhibition has been shown to be protective in OVA- and bleomycin models and the relevant papers quoted.

2. There should also be at some point in the Discussion a mention of the limitations of the models used – many, many compounds giving positive results in these models have failed to translate to therapeutic efficacy in man. A relevant example is mTOR inhibition, which reduced bleomycin-induced fibrosis in mouse but if anything worsened outcomes in human IPF. In addition Line 227 – bleomycin does NOT mimic human pulmonary fibrosis although it is used as a model – please modify this statement.

3. Line 70 – 'lung fibrosis' should be changed to 'airway fibrosis' or 'peri-bronchial fibrosis' as this is airway remodelling that does not affect the lung parenchyma

4. Line 74 – reference 6 barely mentions PI3K inhibitors and only in conjunction with mTOR inhibition - an alternative reference should be used eg Hsu HS et al. Sci Rep. 2017 Oct 27;7(1):14272.

5 Line 158 – does the narrow therapeutic window for this compound pose problems for translation to human studies? Please could this be discussed in the 'Discussion' section. Does the inclusion of tyloxapol as vehicle relate to improving solubility or to improving distribution in the lung? Please clarify in text.

6. Line 175-176 – ‘inhaled CL27c reduced this leukocyte recruitment, though to a lesser extent than dexamethasone (Fig. 2D) – 2D shows only representative histological images – was this data quantified to support the statement and if so how?
7. Similarly in Figure 4B/Figure S6B – what is the ‘Inflammation score’ – how is it derived. There is no description or reference to explain in the methods or supplementary methods.
8. I think Figure S6B should be included in the main figure 5
9. line 360-1 ‘all studies were conducted by RMB – Merck Serono concerning animal care approved by the Official RMB Veterinarian’ - this statement does not make sense to me, please clarify, including what ‘RMB’ stands fo
- 10 Line 538-9 – legend suggests ROS-production was induced by fMLP only but the methods suggest LPS-priming was used – if the latter please include in the legend as priming may entrain additional PI3K isoforms.
11. Line 541 – please state concentration of dexamethasone used
12. Line 543 please supply concentrations of CXCL12 and C5a used
13. Line 545 – please clarify the amount of insulin, route of administration and time of analysis
14. Figure 1E – y axis label – what does ‘f.i.’ indicate?
15. Lines 550-52 – please clarify precise timing of sacrifice/analysis of BALF or tissue on the schedule shown in 2A, also for Figures 3A, 5A and 6A; alternatively make the precise interval clear in the methods
16. Line 552 and line 565 and line 589– Dexa 3 mg/kg i.e. – should this not be x mg/kg i.p?
17. Line 565 – should be ‘Dexamethasone (Dexa)’
18. Line 573 – since CL27c is given together with OVA I would change ‘restores’ to ‘maintains’. Likewise line 577 – change ‘therapeutic’ to ‘preventative’ and in line 581 change ‘recovered’ to ‘maintained’
- 19 line 575 – the appearances are of inflammation not lung injury – please rephrase as ‘lung injury’ has specific connotations.
20. Line 583-4 and line line620-21 – please indicate which test was applied to precisely which data set, not just state either was used.
21. Line 588-9 – should read Dexamethasone (Dexa) 5mg/kg i.p
22. Line 590 – should read ‘alone or with Dexamethasone’
23. line 591 – counts not count
- 24 line 593 – please change ‘association’ to ‘combination’
25. Line line 594 – change ‘recovered’ to maintained

26. Line 599 – change ‘milder disease’ to ‘moderate’ protocol

27. Line 600 – specify route of bleomycin administration

28. line 610 and also 623– why is this ‘a more severe disease protocol’? It is the same protocol as for Figure 6 and should again be labeled as ‘moderate protocol of bleomycin-induced lung fibrosis’. The severe protocol led to 100% death and is only shown in the supplementary figures

Supplementary Material

Line 84 – should be ‘Materials and Methods’

Line 107 – please define ‘CFA’

Line 160-61 – ‘Negative feedback regulation of Rac in leukocytes from mice expressing a constitutively active phosphatidylinositol 3-kinase gamma}. This statement makes no sense in the context it is used, and I think is a cut and paste error – need to describe the chemotaxis conditions.

Line 234-40 – please make it clear how the values presented in Figure 4E were derived and what exactly the y axis label of this figure relates to.

Reviewer #1 (Remarks to the Author):

The studies by Campa and collaborators analyze the pharmacokinetics and efficacy of C27c, an inhaled pan-PI3K inhibitor, in preclinical models of asthma and pulmonary fibrosis. Pan-PI3kinase inhibitors have been approved for some indications; however there is toxicity with systemic and chronic administration. C27c has potential advantages to diminish toxicity since it is a cell permeable pro-drug and can be administered by inhalation. Since previous studies have shown activation of the PI3K pathway in chronic lung diseases such as asthma and fibrosis, the studies are relevant to demonstrate the potential efficacy of this new compound. Despite these translational advantages there are some concerns that the authors need to address.

We thank the referees for having appreciated our work and for having given us the possibility to improve the clarity and strength of our message.

Major concerns.

1. How is PI3K/Akt activation in animal models treated with C27c? Total levels and IHC analysis can also be useful to define cells with effective inhibition of this pathway during disease.

We thank the referee for this significant critique, also raised by referee #2. In the revised version of the paper, we included the suggested IHC and a more quantitative western blot of lung extracts derived from the chronic asthma model. As shown in Fig. S4 B, C and S12 C, D, pAkt can be detected in the areas of either inflammatory cell recruitment (asthma model) or fibrotic remodeling (Bleo model). Conversely, in tissues from CL27-treated mice pAkt is significantly blunted, clearly indicating that CL27c engages its targets in the models of chronic asthma and bleomycin-induced fibrosis.

2. PI3K inhibition has been related to immune toxicity by reduction of T regulatory cells. Since T regs can have a role in airway remodeling and they have been associated with proliferation and Th2 cytokine release in response to allergens, it would be important to define if C27c causes changes in T regs populations in the asthma lung models and if cytokines like TGF- β and IL-10 levels were modified by the compound.

This is an interesting observation, particularly in view of the notion that mice expressing a catalytically inactive PI3K δ have less Tregs^{1,2}. Our new experiments with CL27c show that, in *in vitro* differentiated Tregs, such a toxic effect is reached only at very high concentration (new Supplementary Figure 6A-C), hard to attain in our *in vivo* models. Therefore, CL27c treatment is insufficient to block 100% of PI3K δ *in vivo* as obtained by genetic modification of the *Pik3cd* gene. Consistent with this view, in mice with OVA-elicited allergic asthma, CL27c is able to induce an increase in the production of IL-10 in the lung homogenate (Supplementary Figure 6D). Although IL-10 is produced by a plethora of cell types, Tregs are major producers. Therefore, our findings *in vitro* and *in vivo* strongly converge to no significant changes in Treg

levels after CL27c treatment. Conversely, CL27c significantly reduces TGF- β expression in the lung homogenate from OVA-challenged mice (Supplementary Figure 6E). TGF- β can promote the generation of acquired/induced Treg (aTreg/iTreg) that develop peripherally in response to TGF- β ³. However, while CL27c might disturb the local differentiation of Tregs within the respiratory tract, CL27c is certainly not altering the production of natural Treg (nTreg) from the thymus, which can normally colonize the lungs. Importantly, TGF- β also initiates lung structural remodeling and finding that CL27c reduced TGF- β further supports our observation that pan-PI3K inhibition reduces fibrosis.

3. C27c studies in the ova sensitization models show significant diminution of Th2 mediated responses. However, a key role of Th1 responses have been also recognized in the pathobiology of asthma, particularly in severe types of asthma that is not responsive to corticosteroids so levels of key cytokines as IFN-g is important to define in these models. In addition, C27c was not able to control AHR and IFN-g has been found to be critical to control airway hyperresponsiveness in models of severe asthma (Raundhal et al JCI 2015).

We agree with the referee that it is important to test IFN- γ levels in the model of GC-resistant asthma and we thus measured this cytokine in lungs of control and treated mice. As shown in the new Fig. S9, the OVA-CFA challenge was able to increase IFN- γ abundance. However, neither CL27c nor dexamethasone nor the combination of the two was able to reduce this IFN- γ upregulation. This is in agreement with the notion that, in respiratory diseases, PI3K targeting spares IFN- γ production and has little effects on Th1 responses⁴. In further agreement and, in line with what observed by the referee, CL27c does not affect AHR in the model of severe, glucocorticoid-resistant asthma. However, in the severe, GC-insensitive end of the disease spectrum, the inflammatory cell profile is altered and is characterized by an increased influx of neutrophils, in addition to the presence of eosinophils⁵. The reduction in neutrophil responses to lung recruiting cytokines induced by CL27 might thus explain the protective effects and the re-sensitization to GC triggered by CL27.

4. The authors in the discussion have an erroneous interpretation of the pathogenesis of IPF “Development of pulmonary fibrosis is a consequence of tissue accumulation of activated effector immune cells in the lung”. The most accepted theory of the pathogenesis of IPF is the injury of epithelial cells with activation of profibrotic responses that causes disrepair and accumulation of fibroblasts. Although immune responses can be important to modulate fibrotic responses in IPF, the main pathogenic mechanism is not the accumulation of immune cells.

On contrary, immune responses are important for the development of bleomycin induced lung fibrosis, a model that has several limitations that prevent the direct application to human IPF, as hundreds of studies have shown. Probably, the most relevant model at this point is to use aging mice treated with bleomycin (Jenkins et al AJRCMB 2017, Tashiro et al Front Med 2017).

We are sorry for having given the impression of providing an erroneous interpretation of the pathogenesis of IPF. We changed the discussion to better convey the message that IPF is not an inflammatory disease and stressed the differences between IPF and the bleomycin-induced lung fibrosis. We also addressed the question as to whether CL27c is able to improve fibrotic

remodeling of the lungs independently of its anti-inflammatory activity, following some of the tests suggested by this referee. Although studies in exceptionally old mice might have helped to address this issue^{6, 7}, this turned out to be extremely difficult due to the cost and limited availability of such mice (24 months-old) as well as their intrinsic frailty in response to manipulation. Furthermore, the potential requirement of an infectious insult made this model impossible within the timeframe allotted for this revision. On the contrary, we opted to study the effects of CL27c in fibroblasts from IPF patients and in mice where the bleomycin-induced inflammatory response had resolved, as indicated by the referee the point below.

5. Akt activation has been found to be important for the resistance to apoptosis of lung fibroblasts derived from IPF lungs (Kulasekaran et al AJRCMB 2009). In vitro studies comparing this type of response in lung fibroblasts from donor controls and IPF lungs or the use of human lung slices will bring relevance to the translational potential of this drug. Additional studies using the drug between day 14-21 post bleomycin can also show that the drug is effective not only to control immune responses but also the persistence of fibrosis that is a key component of the disrepair in IPF.

We thank the referee for these insightful suggestions. As requested, we now tested responses to CL27c in human fibroblasts derived from healthy and IPF donors as well as in BLEO-treated mice, starting treatment between days 14-21 (we chose day 17). We found that CL27c is able to significantly reduce phosphorylation of Akt in response to serum plus TGF- β 1 (Fig. S12A), indicating that indeed this treatment can block a key survival signal. Next, we further tested the effects of CL27c in cell-based assays using fibroblasts from healthy as well as IPF patients and found that PI3K signaling blockade significantly reduces cell survival (Fig. S12B). In addition, we seriously took into consideration the request to test CL27c in mice with advanced BLEO-induced pulmonary fibrosis. As shown in the new Fig. S14, treatment with CL27c inhalation for 10 days, starting at day 17 after BLEO, administration still reduces fibrosis, improves lung function and significantly blunts mortality. Although the effect is less pronounced than starting CL27c earlier, these data show that PI3K targeting has an antifibrotic effect independent of its anti-inflammatory activity. Altogether, our observations indicate that PI3K signaling branches on different profibrotic responses and corroborate the conclusive hypothesis reported by Kulasekaran et al.⁸ that “therapy targeting the convergence of downstream signaling pathways common to fibrogenic mediators, rather than targeting a single mediator, may provide improved clinical efficacy for the treatment of pulmonary fibrosis”.

Reviewer #2 (Remarks to the Author):

Campa et al have studied the PK/PD characteristics of a novel inhaled pro-drug (CL27) that inhibits Class 1 PI3Ks, and then demonstrated efficacy of this compound in murine models of OVA-induced acute, chronic and steroid-resistant asthma and bleomycin-induced fibrotic lung disease. The benefits of PI3K inhibition in these models are not novel; the novelty of the manuscript lies in method of active drug delivery, which holds great promise for future therapeutic intervention with restricted toxicity. Whilst I believe this is an exciting and important advance that would be of interest to the journal readership there are some issues that need to be addressed. Overall the statistical analysis seems valid, with a couple of requests for clarification given below.

Major Issues

1. My main concern is that target engagement has not been demonstrated in the relevant models, only following ip-insulin treatment (which is not a pro-inflammatory stimulus). The authors need to show data indicating the impact of CL27c on lung and systemic targets on PI3K activation in at least one of the asthma models and in the bleomycin model.

We thank the referee for this important hint, also suggested by referee #1. In the revised version of the paper, we included the suggested IHC (Fig. S4C and S12D) and a more quantitative western blot of lung extracts (Fig. S4B and S12C) derived from both OVA and BLEO-treated mice. As shown, the pAkt signal is significantly dampened by CL27c administration in both western blots from lung extracts and immunohistochemistry from lung sections. Therefore, CL27c significantly engages its targets in the lungs of mice developing chronic asthma as well as bleomycin-induced fibrosis. Conversely, other organs and potential systemic targets - even in proximity of lungs, such as the heart (see Figure 1 below) - did not show any pAkt signal with or without CL27c, in agreement with OVA and BLEO specifically affecting the airways.

Figure 1: Heart sections from either OVA-induced asthma model or bleomycin model untreated (left panels) or treated with CL27. Paraffin-embedded hearts were sliced and stained with anti-pAkt antibody according to the protocol reported in the Methods section.

2. Figure 1A suggests inhibition of PI3K activity to baseline at 0.3µm CL27c in 'CLL cells', yet full functional inhibition in neutrophils and PBMCs seems to require 10µm. To better relate to the functional data, and also to the models used subsequently, the authors should demonstrate the concentration-dependence of inhibition of AKT phosphorylation in more relevant target cells (eg neutrophils and PBMCs) and should quantify effects by densitometry.

We thank the referee for this important observation. We now provide an extensive validation of EC₅₀ of PI3K pathway inhibition in different leukocyte populations stimulated to trigger different PI3K isoforms, as shown in the cell-based assays reported in Figure 1. In the new Fig. 1A and S3A-D, we now provide pAkt dose responses not only in CLL cells but also in human PBMC as well as murine neutrophils and BMDMs. As correctly noted by the referee, EC₅₀ in CLL cells was lower than for neutrophils stimulated with a GPCR agonist (Fig. 1A and B). This is easily explained by the fact that such stimulation in neutrophils depends on PI3Kγ, the isoform against which CL27c shows the lowest IC₅₀. The functional data in Fig. 1B is now perfectly in line with the EC₅₀ for pAkt in isolated mouse neutrophils. In addition, we analyzed the dose/response to CL27c in human PBMCs as requested. In line with a stronger effect of CL27c against PI3Kδ than PI3Kγ⁹ (please also see our reply to point #3), results with CLL cells, shown in Fig. 1A, are in line with what observed in human PBMC stimulated to activate PI3Kδ (Fig. S3C). Consistent with these observations, PBMC show a sensitivity to pAkt inhibition (Fig. 3C) compatible with results of the proliferation assay (Figure 1C).

3. Suppl Fig S2/Table S4. As far as I can see, the inhibitory profile of CL27 is only given for PI3Kα, the isoform likely least likely to be relevant to the models employed. Data for all other Class I and also Class 2/3 PI3Ks, plus AKT and mTOR should be included in these tables. If this is not possible for some reason, perhaps the IC50s of CL27c for all of these kinases could be presented in a separate table?

The IC₅₀s of CL27 for all class I (PI3Kα IC₅₀=18 nM; PI3Kδ IC₅₀=19 nM; PI3Kβ IC₅₀=59 nM; PI3Kγ IC₅₀= 186 nM) as well as class II and III have been already published⁹. Please refer to this study for the detailed characterization of CL27 compounds. On the contrary, the profiling of CL27e against a subset of the kinome in Fig. S2 has not been published before.

4. The figure legends are not well written – there are several typos and omissions (see below). In some cases the authors just recapitulate a summary statement of the results without indicating how the experiment was undertaken, and the latter needs to be conveyed in brief. Please could the authors re-read the legends carefully and optimise them.

We apologize for the poorly written figure legends. We now re-wrote all the captions to follow the Referee's indications and the journal style.

5. Importantly related to the above please clarify the doses of CL27c used in each model. In the main text and the methods section it states 2mg/ml/mouse was used in all cases, and this dose is quoted in the legend for Figure 2, but in the legends for Figures 3, 4, 5 and 7 it says 0.2mg/ml/mouse was used, whilst in Figure 6 the dose is given as '2mg/kg'. It is very important that the manuscript is internally consistent on this issue.

We apologize for the confusion due to repeated cut and paste of a typo. The dose used in all models was 2mg/ml. Figure legends and text have been amended to more clearly state this point.

Minor Issues and Typos

1. It should be clearly stated in the Introduction that PI3K inhibition has been shown to be protective in OVA- and bleomycin models and the relevant papers quoted.

The introduction now stresses this point.

2. There should also be at some point in the Discussion a mention of the limitations of the models used – many, many compounds giving positive results in these models have failed to translate to therapeutic efficacy in man. A relevant example is mTOR inhibition, which reduced bleomycin-induced fibrosis in mouse but if anything worsened outcomes in human IPF. In addition Line 227 – bleomycin does NOT mimic human pulmonary fibrosis although it is used as a model – please modify this statement.

We accordingly modified the end of the revised discussion. The paper by Malouf et al., reporting the negative results of the clinical trial with everolimus in IPF, is cited in new line 325.

3. Line 70 – 'lung fibrosis' should be changed to 'airway fibrosis' or 'peri-bronchial fibrosis' as this is airway remodelling that does not affect the lung parenchyma

Text has been accordingly modified. Please see line 70.

4. Line 74 – reference 6 barely mentions PI3K inhibitors and only in conjunction with mTOR inhibition - an alternative reference should be used eg Hsu HS et al. Sci Rep. 2017 Oct 27;7(1):14272.

The old reference 6 has been removed and substituted with the suggested paper by Hsu et al.

5 Line 158 – dose the narrow therapeutic window for this compound pose problems for translation to human studies? Please could this be discussed in the 'Discussion' section. Does the inclusion of tyloxapol as vehicle relate to improving solubility or to improving distribution in the lung? Please clarify in text.

This issue has been included in the discussion. Please see lines 347. Tyloxapol was added to improve distribution in the lungs as well as solubility of CL27c, as now indicated in Material and Methods.

6. Line 175-176 – 'inhaled CL27c reduced this leukocyte recruitment, though to a lesser extent than dexamethasone (Fig. 2D) – 2D shows only representative histological images – was this data quantified to support the statement and if so how?

As a precise quantification of CD18-positive cells was at this stage not possible, we decided to remove the cited statement. We thank the referee for this suggestion that helps to improve the quality of our message.

7. *Similarly in Figure 4B/Figure S6B – what is the ‘Inflammation score’ – how is it derived. There is no description or reference to explain in the methods or supplementary methods.*

Details have been added in the Supplementary Materials and Methods section, describing how the inflammation score was calculated.

8. *I think Figure S6B should be included in the main figure 5*

Done as requested.

9. *line 360-1 ‘all studies were conducted by RMB – Merck Serono concerning animal care approved by the Official RMB Veterinarian’ - this statement does not make sense to me, please clarify, including what ‘RMB’ stands for*

We are sorry for the inconsistent sentence referring to the name of the Merck Serono outpost in Italy (RBM). RBM=Ricerche Bio-Mediche (BioMedical Research); we thank the referee for spotting the typo. A clearer text is now provided.

10 *Line 538-9 – legend suggests ROS-production was induced by fMLP only but the methods suggest LPS-priming was used – if the latter please include in the legend as priming may entrain additional PI3K isoforms.*

As correctly stated in the Methods, neutrophils have been primed with LPS following the protocol used by Hirsch et al., Science 2000¹⁰. As reported, LPS priming in mouse neutrophils induces the upregulation of the fMLP receptor. The kinetic of fMLP response within few minutes is PI3Kgamma selective, as this is lost in KO neutrophils. Engagement of other PI3K isoforms occurs at a later time¹¹. In compliance with the reviewer’s request, we indicated the LPS-priming in the figure legend as well.

11. *Line 541 – please state concentration of dexamethasone used*

Dexamethasone was 5 mg/kg via i.p. injection as now indicated.

12. *Line 543 please supply concentrations of CXCL12 and C5a used*

CXCL12 and C5a were 100nM and 50 nM, respectively. The text has been accordingly amended.

13. *Line 545 – please clarify the amount of insulin, route of administration and time of analysis*

Insulin was 1UI /kg in PBS solution I.P. injection.

14. *Figure 1E – y axis label – what does ‘f.i’ indicate?*

This represents Fold Induction relative to the baseline. The figure legend has been accordingly amended.

15. Lines 550-52 – please clarify precise timing of sacrifice/analysis of BALF or tissue on the schedule shown in 2A, also for Figures 3A, 5A and 6A; alternatively make the precise interval clear in the methods

We now added all the precise intervals in the Methods section. The figures have been updated to eliminate the temporal arrows.

16. Line 552 and line 565 and line 589– Dexa 3 mg/kg i.e. – should this not be x mg/kg i.p?

We thank the referee for having spotted this typo that we now accordingly corrected.

17. Line 565 – should be ‘Dexamethasone (Dexa)’

Accordingly changed.

18. Line 573 – since CL27c is given together with OVA I would change ‘restores’ to ‘maintains’. Likewise line 577 – change ‘therapeutic’ to ‘preventative’ and in line 581 change ‘recovered’ to ‘maintained’

We thank the referee for allowing us to improve the precision of our message. Text has been accordingly changed.

19 line 575 – the appearances are of inflammation not lung injury – please rephrase as ‘lung injury’ has specific connotations.

The figure legend has been accordingly changed.

20. Line 583-4 and line line620-21 – please indicate which test was applied to precisely which data set, not just state either was used.

Done as indicated.

21. Line 588-9 – should read Dexamethasone (Dexa) 5mg/kg i.p

Corrected as indicated

22. Line 590 – should read ‘alone or with Dexamethasone’

Corrected as indicated

23. line 591 – counts not count

Corrected as indicated

24 line 593 – please change ‘association’ to ‘combination’

Corrected as indicated

25. Line line 594 – change ‘recovered’ to maintained

Corrected as indicated

26. Line 599 – change ‘milder disease’ to ‘moderate’ protocol

Corrected as indicated

27. Line 600 – specify route of bleomycin administration

Bleomycin was administered through the intranasal route, as now indicated.

28. line 610 and also 623– why is this ‘a more severe disease protocol’? It is the same protocol as for Figure 6 and should again be labeled as ‘moderate protocol of bleomycin-induced lung fibrosis’. The severe protocol led to 100% death and is only shown in the supplementary figures

We are sorry for this confusing inconsistency. The referee is right; this is not a “more severe disease protocol” as it is the same moderate protocol defined in Figure 5. The manuscript has been now accordingly modified.

Supplementary Material

Line 84 – should be ‘Materials and Methods’

Done

Line 107 – please define ‘CFA’

Complete Freund’s Adjuvant (CFA). Done as requested.

Line 160-61 – ‘Negative feedback regulation of Rac in leukocytes from mice expressing a constitutively active phosphatidylinositol 3-kinase gamma}. This statement makes no sense in the context it is used, and I think is a cut and paste error – need to describe the chemotaxis conditions.

We are sorry for the cut and paste mistake. We added more indications on chemotaxis conditions.

Line 234-40 – please make it clear how the values presented in Figure 4E were derived and what exactly the y axis label of this figure relates to

Text has been amended to better clarify the procedure

References

1. Ali K, *et al.* Inactivation of PI(3)K p110delta breaks regulatory T-cell-mediated immune tolerance to cancer. *Nature* **510**, 407-411 (2014).
2. Hirsch E, Novelli F. Cancer: natural-born killers unleashed. *Nature* **510**, 342-343 (2014).
3. Al-Alawi M, Hassan T, Chotirmall SH. Transforming growth factor beta and severe asthma: a perfect storm. *Respiratory medicine* **108**, 1409-1423 (2014).
4. Medina-Tato DA, Ward SG, Watson ML. Phosphoinositide 3-kinase signalling in lung disease: leucocytes and beyond. *Immunology* **121**, 448-461 (2007).

5. DeJager L, *et al.* Neutralizing TNF α restores glucocorticoid sensitivity in a mouse model of neutrophilic airway inflammation. *Mucosal Immunol* **8**, 1212-1225 (2015).
6. Jenkins RG, *et al.* An Official American Thoracic Society Workshop Report: Use of Animal Models for the Preclinical Assessment of Potential Therapies for Pulmonary Fibrosis. *Am J Respir Cell Mol Biol* **56**, 667-679 (2017).
7. Tashiro J, *et al.* Exploring Animal Models That Resemble Idiopathic Pulmonary Fibrosis. *Frontiers in medicine* **4**, 118 (2017).
8. Kulasekaran P, Scavone CA, Rogers DS, Arenberg DA, Thannickal VJ, Horowitz JC. Endothelin-1 and transforming growth factor- β 1 independently induce fibroblast resistance to apoptosis via AKT activation. *Am J Respir Cell Mol Biol* **41**, 484-493 (2009).
9. Pirali T, *et al.* Identification of a Potent Phosphoinositide 3-Kinase Pan Inhibitor Displaying a Strategic Carboxylic Acid Group and Development of Its Prodrugs. *ChemMedChem*, (2017).
10. Hirsch E, *et al.* Central role for G protein-coupled phosphoinositide 3-kinase γ in inflammation. *Science* **287**, 1049-1053 (2000).
11. Condliffe AM, *et al.* Sequential activation of class IB and class IA PI3K is important for the primed respiratory burst of human but not murine neutrophils. *Blood* **106**, 1432-1440 (2005).

Reviewer #1 (Remarks to the Author):

The authors have answered several of the previous comments. There are still some concerns that need to be addressed.

1. The inflammatory score to evaluate the asthma models is highly subjective without semi or quantitative parameters. It does not represent a valid comparative method to compare severity of inflammation.
2. Please list the source of IPF lung fibroblasts, the number of lines and the passage of the cells used in the experiments.
3. Interestingly, there is no correlation between the elevated signal of p-Akt in lungs of bleo treated mice and the lower p-Akt levels in IPF lung fibroblasts compared with healthy fibroblasts in panel S13c. This data contradicts the relevance of the Akt pathway in the activated phenotype of the IPF lung fibroblasts. In contrary, high levels of pAkt have been found in IPF lung epithelial cells associated with low levels of PTEN and loss of epithelial integrity (Miyoshi et al 2013) and that might be a better cell type to analyze the therapeutic effect of p-Akt inhibition.

Reviewer #2 (Remarks to the Author):

The authors have greatly improved the manuscript, by adding additional data and by correcting a number of typographic errors in particular in the figure legends. I believe the body of data as presented warrants publication. There are some minor issues in phrasing that I think relate to English not being the first language of the authors but these do not affect comprehension of the manuscript/

Reviewer #1 (Remarks to the Author):

The authors have answered several of the previous comments. There are still some concerns that need to be addressed.

1. The inflammatory score to evaluate the asthma models is highly subjective without semi or quantitative parameters. It does not represent a valid comparative method to compare severity of inflammation.

We understand the referee's concern but we provided a description of the semi-quantitative parameters used in the extended method section. We think that any potential bias in comparing severity of inflammation has been called off as follows:

- 1) As requested by Referee #2, minor point #7, we have already provided a detailed supplementary methods section (Lung histopathology) indicating the semi-quantitative scale used to determine the score.
- 2) As stated in the Methods, we had double-blinded pathologists assessing the sections
- 3) We do not base our conclusions on the inflammatory score alone (please refer to Figure 2; Fig. 3, Fig. 4d,e,f; Fig. 5). We think that Panels 4B and 5D do not weaken our much deeper quantitative analysis.
- 4) Last but not least, we had to include the pathological score in main figures to comply with the explicit requests from Referee #2 (please see Minor Issues #8 "I think Figure S6B should be included in the main figure 5"). Now Referee #1 opposes what required by Referee #2.

Therefore, we believe that panels 4B and 5D should remain in the main figures as a support to the other purely quantitative observations. This is in line with a large number of other papers very recently published in Nature Communications and showing histology score in main figures [e.g.: Tye et al., Nature Communications volume 9, Article number: 3728 (2018)].

2. Please list the source of IPF lung fibroblasts, the number of lines and the passage of the cells used in the experiments.

We thank the referee for this remark that allowed us to better label fibroblast cell lines used in this study. The fibroblasts are the following:

- 1) NHLF – Primary Human Lung Fibroblasts (CC-2512; Lonza); passage 3-6 (assured for experimental use for 15 population doublings)
- 2) DHLF-IPF – Primary Diseased Human Lung Fibroblasts, Idiopathic Pulmonary Fibrosis (CC-7231; Lonza); passage 2-3 (assured for experimental use for 3 population doublings)
- 3) CCD-16Lu Healthy male (35 years old); (ATCC® CCL-204™); passage 3-6.
- 4) LL 97A (AlMy) IPF male (48 years old); (ATCC® CCL-191™); passage 3-6.

These pieces of information are now present in the Supplementary Materials and Methods and in the new Figure S13C.

3. Interestingly, there is no correlation between the elevated signal of p-Akt in lungs of bleo treated mice and the lower p-Akt levels in IPF lung fibroblasts compared with healthy fibroblasts in panel S13c. This data contradicts the relevance of the Akt pathway in the activated phenotype of the IPF lung fibroblasts. In contrary, high levels of pAkt have been found in IPF lung epithelial cells associated with low levels of PTEN and loss of epithelial integrity (Miyoshi et al 2013) and that might be a better cell type to analyze the therapeutic effect of p-Akt inhibition.

We agree with the referee that results in old panel S13c are not suitable to show increased PI3K pathway activation in IPF. By displaying the response of healthy and IPF fibroblasts in different blots we invalidated the possibility of comparing the level of PI3K pathway activation in health and disease. In addition, in the old figure we showed stimulation of the PI3K/Akt pathway by TGF/beta1, a weak PI3K activator, unsuitable to show subtle differences due, for example, to PTEN downregulation. For this reason, to better comply with the referee's suggestion we repeated pAkt measurements in different conditions and with correct samples in the same gel. This time, we triggered PI3K/Akt pathway activation by serum-containing culture medium, as in the paper by Miyoshi et al., 2013. In addition, we loaded samples of healthy and IPF fibroblasts on the same gel. This allowed to observe that, in line with what suggested by the referee, Akt phosphorylation level is higher in IPF fibroblasts of two independent sources than in their respective controls. This perfectly matches what observed in mice where fibrosis is accompanied by pAkt elevation. Remarkably, treatment with CL27c restored, if not totally blocked, pAkt in IPF fibroblasts. This fits the idea that activation of the PI3K/Akt pathway is a hallmark of IPF and that PI3K inhibitors can block a key process leading to disease.

Reviewer #2 (Remarks to the Author):

The authors have greatly improved the manuscript, by adding additional data and by correcting a number of typographic errors in particular in the figure legends. I believe the body of data as presented warrants publication. There are some minor issues in phrasing that I think relate to English not being the first language of the authors but these do not affect comprehension of the manuscript/

We thank the referee for supporting the publication of our results and apologize for any remaining language use issues.

Reviewer #1 (Remarks to the Author)

The authors have answered several of the previous comments. There are still some concerns that need to be addressed.

1. The inflammatory score to evaluate the asthma models is highly subjective without semi or quantitative parameters. It does not represent a valid comparative method to compare severity of inflammation.
2. Please list the source of IPF lung fibroblasts, the number of lines and the passage of the cells used in the experiments.
3. Interestingly, there is no correlation between the elevated signal of p-Akt in lungs of bleo treated mice and the lower p-Akt levels in IPF lung fibroblasts compared with healthy fibroblasts in panel S13c. This data contradicts the relevance of the Akt pathway in the activated phenotype of the IPF lung fibroblasts. In contrary, high levels of pAkt have been found in IPF lung epithelial cells associated with low levels of PTEN and loss of epithelial integrity (Miyoshi et al 2013) and that might be a better cell type to analyze the therapeutic effect of p-Akt inhibition.

Reviewer #2 (Remarks to the Author)

The authors have greatly improved the manuscript, by adding additional data and by correcting a number of typographic errors in particular in the figure legends. I believe the body of data as presented warrants publication. There are some minor issues in phrasing that I think relate to English not being the first language of the authors but these do not affect comprehension of the manuscript/